🔓 | **Open Peer Review** | Host-Microbial Interactions | Research Article

# Altered microRNA expression correlates with reduced TLR2/4-dependent periodontal inflammation and bone resorption induced by polymicrobial infection

Syam Jeepipalli,[1] Parvathi Gurusamy,[1] Ana Rafaela Luz Martins,[1] Eduardo Colella,[1] Sandhya R. Nadakuditi,[1] Tushar Desaraju,[1] Ashitha Yada,[1] Jennifer Onime,[1] John T. William,[2] Indraneel Bhattacharyya,[3] Edward K. L. Chan,[2] L. Kesavalu[1,2]

**ABSTRACT**  Periodontitis (PD) is a polymicrobial dysbiotic immuno-inflammatory disease. Toll-like receptors (TLRs) are present on gingival epithelial cells and recognize pathogen-associated molecular patterns on pathogenic bacteria, inducing the secretion of proinflammatory cytokines and initiating innate and adaptive antigen-specific immune responses to eradicate the invading microbes. Since PD is a chronic inflammatory disease, TLR2/TLR4 play a vital role in disease pathogenesis and in maintaining the periodontium during health. Many factors modulate the TLR-mediated signaling pathway, including specific microRNAs (miRNAs). The present study was designed to characterize the function of TLR2/4 signaling in the miRNA profile after polybacterial infection with *Streptococcus gordonii, Fusobacterium nucleatum, Porphyromonas gingivalis, Treponema denticola,* and *Tannerella forsythia* in C57BL6/J wild-type, global knockout strains of TLR2$^{-/-}$, and TLR4$^{-/-}$ mice ($n$ = 16/group) using RT-qPCR. The selection of 15 dominant miRNAs for RT-qPCR analysis was based on prior NanoString global miRNA expression profiling in response to polymicrobial and monobacterial infection. Polybacterial infections established gingival colonization in wild-type, TLR2$^{-/-}$, and TLR4$^{-/-}$ mice with the induction of bacterial-specific IgG. A significant reduction in alveolar bone resorption and gingival inflammation was observed in the mandibles of TLR2/4$^{-/-}$ mice compared to C57BL6/J wild-type mice ($P$ < 0.0001). Periodontal bacteria disseminated from gingival tissue to multiple organs in wild-type and TLR2$^{-/-}$ mice (heart, lungs, brain, and kidneys) and were limited to the heart (*F. nucleatum*), lungs (*P. gingivalis*), and kidneys (*T. forsythia*) in TLR4$^{-/-}$ mice. The diagnostic potential of miRNAs was assessed by receiver operating characteristic curves. Among the 15 miRNAs, three were upregulated in C57BL6/J wild-type mice, two in TLR2$^{-/-}$, and seven in TLR4$^{-/-}$ mice. Notably, the anti-inflammatory miR-146a-5p was consistently upregulated in wild-type and TLR2$^{-/-}$ mice but not in TLR4$^{-/-}$ mice. Additionally, miR-15a-5p was upregulated in wild-type and TLR2$^{-/-}$ mice. let-7c-5p was upregulated in TLR4$^{-/-}$ mice and downregulated in wild-type mice. The upregulated miRNAs (miR-146a, miR-15a-5p, and let-7c-5p) and downregulated miRNAs could be candidate biomarkers for oral bacteria and TLRs-mediated induction of periodontitis if they are confirmed in large-scale human studies. Multi-species oral bacterial infection alters the TLR2/4 signaling pathways by modulating the expression of several potential biomarker miRNAs in the periodontium.

**IMPORTANCE**  Periodontitis is the most prevalent chronic immuno-infectious multispecies dysbiotic disease of the oral cavity. The Toll-like receptors (TLRs) provide the first line of defense, one of the best-characterized pathogen detection systems, and play a vital role in recognizing multiple microbial products. Multispecies infection with periodontal bacteria *Streptococcus gordonii*, *Fusobacterium nucleatum*, *Porphyromonas gingivalis*, *Treponema denticola*, and *Tannerella forsythia* induced gingival inflammation, alveolar

**Peer Reviewer** Kamoru A. Adedokun, Roswell Park Comprehensive Cancer Center, Buffalo, New York, USA

Address correspondence to L. Kesavalu, kesavalu@dental.ufl.edu.

The authors declare no conflict of interest.

See the funding table on p. 20.

bone resorption (ABR), and microRNA (miRNA) expression in the C57BL6/J wild-type mice, whereas infection did not significantly increase ABR in the TLR2/4 deficient mice. Among the 15 miRNAs investigated, miR-146a-5p and miR-15a-5p were upregulated in wild-type and TLR2[-/-] mice, and miR-146a-5p, miR-30c-5p, and let-7c-5p were upregulated in the TLR4[-/-] mice compared to sham-infected controls. Notably, the inflammatory miRNA miR-146a-5p was uniquely upregulated among the wild-type and TLR2[-/-] mice but not in the TLR4[-/-] mice. The upregulated miRNAs (miR-146a, miR-15-a-5p, let-7c-5p) and downregulated miRNAs could be candidate biomarkers for oral bacteria and TLRs-mediated induction of periodontitis if they are confirmed in large-scale human studies.

KEYWORDS    Toll-like receptors, polymicrobial infection, oral bacteria, microRNAs, periodontal disease

Periodontitis (PD) is an immuno-inflammatory polymicrobial dysbiotic disease characterized by complex subgingival plaques and leads to the inflammatory destruction of the supporting tissues, including gingival tissue, periodontal ligament, and alveolar bone. Several studies report that *Porphyromonas gingivalis*, *Tannerella forsythia*, *Treponema denticola,* and *Fusobacterium nucleatum* (co-aggregating bacteria) are more frequently identified, synergistically interact with each other, and in higher numbers in adult periodontitis compared to healthy individuals. They are also positively correlated with pocket depth and bleeding on probing, which are measures of periodontal tissue destruction (1).

MicroRNAs (miRNAs) are small (20–24 nucleotides long) noncoding RNA molecules that play a critical role in regulating gene expression by directly binding to their targeted 3′ untranslated regions (2, 3). Circulating miRNAs have proven to be diagnostic markers in cancer (miR-15, miR-339, miR-375, and miR-133a), cardiovascular diseases (miR-133a), non-alcoholic fatty liver disease (miR-34a and miR-375), neurodegenerative diseases (miR-146a), and the prognostic miR-423-5p in acute heart failure (4, 5). Certain miRNAs (miR-15a, miR-30d, miR-146a, and miR-155) are implicated in the pathogenesis of PD (6–9). The periodontal miRNAs expressed in the polybacterial infection (miR-375, miR-690, miR-148a, and mmu-let-7a-5p) are not identical to those in monobacterial infection, such as *P. gingivalis* (miR-133a and miR-22)*, T. denticola* (miR-133a*,* miR-378, and miR-34b-5p)*, T. forsythia* (miR-1902 and miR-720)*, F. nucleatum* (miR-361 and miR-323-3p)*,* and *S. gordonii* (miR-135a and miR-720) (10–15).

Clinical studies have shown that mRNA expression of TLR2 and TLR4 is significantly elevated in gingivitis (16) and in tissues affected by severe periodontitis (17, 18). These results show that TLR2 and TLR4 could play a major role in pathogenesis. It is known, for example, that lipoproteins from *P. gingivalis* (19) and other bacteria are known to activate TLR2 (20), while *P. gingivalis* LPS activates TLR4 (21–23). Also, our report focusing on atherosclerosis in this mouse model shows that polybacterial infections have established gingival colonization and induction of a pathogen-specific IgG immune response in TLR2[-/-] and TLR4[-/-] mice after several weeks of chronic gingival infection (24). TLR2/4 deficiency dampened alveolar bone resorption (ABR) and intrabony defects, indicating the central role of TLR2/4 in polymicrobial infection-induced periodontitis (24). Thus, it is interesting to define the relative contribution of TLR2/4 and the function of their associated dominant miRNAs to the pathogenesis in our polymicrobial PD model. TLR2 is detected in gingival pocket epithelia, gingival fibroblasts, neutrophils, cementum, periodontal ligament fibroblasts (PLFs), osteoclasts, and tissue dendritic cells, whereas TLR4 is predominantly detected in the gingival epithelium, fibroblasts, cementum, PLFs, dendritic cells, osteoblasts, and endothelium (25, 26). TLRs provide the first line of defense and are one of the best-characterized pathogen-detection systems. They play a vital role in recognizing multiple microbial products, including LPS, lipoproteins, peptidoglycan, lipoteichoic acid, and fimbriae. Activation of TLRs triggers the release of proinflammatory mediators, such as TNFα, IL-1β, and IL-8, through the induction of transcription factors NF-*k*B (nuclear factor kappa-light-chain-enhancer of activated

B cells); this signaling pathway is essential for initiating an immune response and driving gingival inflammation, resulting in adaptive immune responses. TLR2 recognizes peptidoglycans, lipoproteins, and lipoarabinomannans from gram-positive bacteria, and TLR4 is activated by the gram-negative bacterial component LPS. In addition, several miRNAs can regulate TLR expression and initiate adaptive immune responses by inducing immune and inflammatory gene expression. TLR2/TLR4 (2,4) are expressed both on immune cells (dendritic cells, macrophages, B cells, and natural killer cells) and in periodontal tissues (27), interact with pathogen-extracellular stimuli Microbe-associated molecular patterns (MAMPs) (28), and play a central role in gene expressions, inflammation, and different stages of periodontitis (17, 29).

Periodontal bacteria surface antigens in gingivitis and periodontitis interact with TLRs and play an essential role in developing adaptive immunity (30, 31). Reciprocal extracellular miRNA vesicle communication between different cell populations facilitates the host response (32). In our previously published study, mice deficient in TLR2 and TLR4 affected both periodontitis and atherosclerosis (24, 33) induced by polymicrobial infections. Identifying the miRNA markers at the early stages of PD is essential to initiate timely interventions that prevent disease complications and preserve oral health. Diagnostic miRNA discovery from the preclinical studies could help clinicians identify the disease severity in human periodontitis, e.g., miR-146a-5p (34, 35). The high expression of miR-146a was observed in many inflammatory diseases, such as osteoarthritis and rheumatoid arthritis (36).

The preclinical *in vivo* studies involving five different bacteria (*S. gordonii, F. nucleatum, P. gingivalis, T. denticola,* and *T. forsythia*) have revealed specific differentially expressed (DE) miRNAs in each infection that were analyzed using NanoString nCounter technology (10–13, 15). These miRNAs exhibit a unique range of expression patterns during polymicrobial infections (10), as well as in five distinct monoinfection (*S. gordonii, P. gingivalis, F. nucleatum, T. denticola,* and *T. forsythia*) (11, 13, 15). With the highest upregulated and downregulated miRNAs showing high fold change, 15 miRNAs were selected for investigation of their presence in C57BL6/J wild type (hereafter termed as wild type), global TLR2 gene knockout mice, and global TLR4 gene knockout mice (Table 2) mandibles during polymicrobial infection. The 15 selected miRNAs are mmu-let-7c-5p, miR-15a-5p, miR-22-5p, miR-30c-5p, miR-34b-5p, miR-133a-3p, miR-146a-5p, miR-323-3p, miR-339-5p, miR-375-3p, miR-361-5p, miR-423-5p, miR-720, miR-155-5p, and miR-132-3p. Reverse-transcription quantitative polymerase chain reaction (RT-qPCR) is an effective method for studying gene expression and measuring the expression level of target genes. In the present study, we aimed to investigate the critical function of these 15 preselected dominant periodontal miRNAs using RT-qPCR in polymicrobial-infected (mimicking the human oral microbiota ecological colonization) wild-type, TLR2$^{-/-}$, and TLR4$^{-/-}$ male and female mice mandibles.

## MATERIALS AND METHODS

### Experimental mice and housing conditions

This experiment used C57BL6/J wild type, TLR2$^{-/-}$ B6.129-Tlr2tm1Kir/J (Strain # 004650), a global knockout for TLR2 receptors (37), and TLR4$^{-/-}$ B6(Cg)-Tlr4tm1.2Karp/J (Strain # 029015) mice, a global knockout for TLR4 receptors, which were purchased from Jackson Laboratory (Bar Harbor, ME, USA). Upon arrival, the mice were allowed a week of acclimation before initiating the infection and sham infection. At the time of infection, we used 9-week-old mice for this study. Throughout the study, mice were housed in a controlled environment with 12 h of dark/light cycles and maintained at a consistent temperature. Mice had access to standard chow and water *ad libitum*. Mice were randomly assigned to the infection and sham infection group for each genotype (wild type, TLR2$^{-/-}$, and TLR4$^{-/-}$). Specific efforts were followed to ensure minimal pain and suffering, and approved isoflurane inhalation anesthesia methods were employed for the mice.

## Grouping the mice

Mice were divided into polybacterial infection groups and sham infection groups. Each group consists of 16 mice (8 male and 8 female). The sample size was determined based on previous studies from Aravindraja et al. (10, 11, 13, 15). Mice in group I (wild-type mice), III (TLR2[-/-] mice), and V (TLR4[-/-] mice) underwent sequential polymicrobial infection cycles (ICs). These infection cycles followed a specific order: *S. gordonii* (*Sg*, 2 ICs), *F. nucleatum* (*Fn*, 2 ICs), and later with *P. gingivalis* (*Pg*), *T. denticola* (*Td*), and *T. forsythia* (*Tf*) (5 ICs). Each infection cycle consisted of 4 days of intraoral bacterial infection in a week, using $2.5 \times 10^8$ cells of each bacterial species suspended in reduced transport fluid (RTF) + 6% CMC (10), while sham-infected Group II (wild type), Group IV (TLR2[-/-]), and Group VI (TLR4[-/-]) mice were mock-infected with RTF + 4% CMC only (Fig. 1A).

## Bacterial culture and oral administration

In this study, the following bacterial strains were used: *S. gordonii* DL1, *F. nucleatum* ATCC 49256, *P. gingivalis* ATCC 53977, *T. denticola* ATCC 35404, and *T. forsythia* ATCC 43037. Standard protocol for bacterial growth culture, harvesting, and cell counting procedures was followed as described by Aravindraja et al. (10, 24). Before bacterial infection, mice were allowed to consume kanamycin water (500 mg/2 L) for 3 days, and the oral cavity was rinsed with 0.12% chlorohexidine gluconate to further reduce bacterial growth (10).

## Molecular detection of bacteria in the gingival swabs and systemic organs

Gingival swabs from each mouse were collected after the fourth day of an infection cycle using a sterile cotton swab, then suspended in Tris-EDTA (TE) buffer and subjected to colony polymerase chain reaction (PCR). Testing was performed in a Bio-Rad thermal cycler (Bio-Rad, Hercules, CA, USA) with the test reaction components, such as master mix (NEB, Ipswich, MA, USA), 16s rRNA-specific forward primer (FP), reverse primer (RP), and the DNA source (oral swabs). Details of each bacterial primer are shown in Table 1. DNA extracted from pure bacterial culture was considered a positive control, and sterile PCR-grade water was used as a negative control. PCR products were run in 1% agarose gel electrophoresis and visualized using UVP GelStudio Touch Imaging System (Analytik Jena US LLC, CA, USA) (10, 24). Bacterial genomic DNA (gDNA) from the organs was harvested following the standard protocol described in the Qiagen DNeasy Blood and Tissue Kit (Qiagen, Germantown, MD, USA). The PCR test was performed as described above.

## Euthanasia and sample collection

Mice were euthanized a week after the 10th infection cycle following the carbon dioxide ($CO_2$) inhalation procedure. Blood was drawn through a cardiac puncture procedure, and the serum was isolated and stored at −20°C to measure immunoglobulin G (IgG) concentration against each bacterium. Additionally, the distal organs (heart, liver, kidney, spleen, lungs, and brain) were harvested and stored at −80°C for further examination. The left maxilla and mandibles were excised and preserved in RNAlater at −80°C for the total RNA extraction. The right maxilla and mandibles were excised and processed for alveolar bone resorption morphometry analysis.

## Serum antibody analysis

Serum isolated from the wild-type mice, TLR2[-/-] mice, and TLR4[-/-] mice was used to analyze the levels of periodontal bacteria (*Sg, Fn, Pg, Td,* and *Tf*). Surface antigen-specific IgG was determined separately by enzyme-linked immunosorbent assay (ELISA) as described by Aravindraja et al. (10, 38). Briefly, the formaldehyde-killed bacteria, such as *F. nucleatum*, *P. gingivalis*, *T. denticola*, and *T. forsythia,* were used for coating antigens to the ELISA plate wells, and diluted serum (100 µL) of each mouse was added to each well in triplicate. Goat anti-mouse-IgG alkaline phosphatase (Sigma-Aldrich, St.

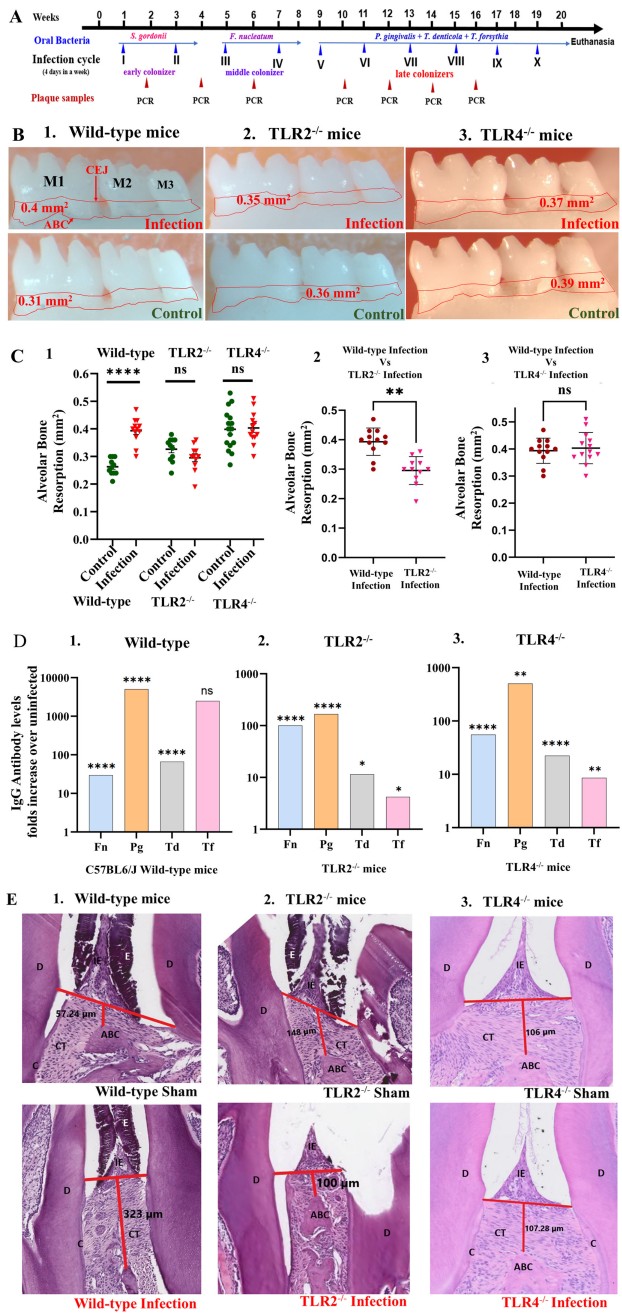

FIG 1 Intraoral ecological time-sequential polymicrobial periodontal infection (ETSPPI) results in alveolar bone resorption. (A) Schematic diagram of the experimental design depicting the ETSPPI infection (4 days per week on every alternate week), plaque sampling for PCR, and euthanasia. (B) Representative images show horizontal ABR (mandible lingual view) of the polybacterial infected and sham-infected mice with the area of ABR outlined from the alveolar bone crest (ABC) to the cementoenamel junction (CEJ). (C) Morphometric analysis of the mandible and maxillary ABR in mice. Two-way ANOVA. C1 represents the ABR in wild type, TLR2[-/-], and TLR4[-/-] mice. A significant increase in ABR was observed in the mandible lingual ($P < 0.0001$) and maxilla buccal ($P < 0.01$) in the wild-type mice (C1). Polymicrobial infection did not show ABR in the mandible lingual, maxilla palatal, and maxilla buccal sides in the TLR2[-/-] mice (C1) and TLR4[-/-] mice (C1) relative to their uninfected control mice. Polymicrobial infection increased the ABR in the wild-type ($P < 0.05$) compared to polymicrobial-infected TLR2[-/-] mice (C2). The polymicrobial infection did not show significant ABR among the wild-type and polymicrobial-infected TLR4[-/-] mice (C3). (D) Bacteria-specific IgG antibody analysis. (D) shows a presentation of IgG expression in fold change

Fig 1 (Continued)

in the infection over sham. The ordinary two-way ANOVA was performed ($n = 16$). (E) Representative hematoxylin and eosin-stained histological mandible tissue sections from polymicrobial-infected and sham-infected wild-type (left), TLR2$^{-/-}$ (middle), and TLR4$^{-/-}$ mice (right) ($n = 5$) showing infiltration of inflammatory cells. The red line indicates vertical ABR from ABC to CEJ. D, dentin; E, enamel matrix; C, cementum; CT, connective tissue; and IE, interdental periodontal epithelium. Bar = 100 μm. Images are shown at 40× magnification.

Louis, MO, USA) was added to each reaction well, incubated, and color developed with *p*-nitrophenyl phosphate (Sigma-Aldrich, St. Louis, MO, USA). Color development was stopped by adding 3 M NaOH, and the yellow color intensity was measured at OD$_{405\ nm}$ using an Epoch microplate spectrophotometer and analyzed in Gen5 software (BioTek, Winooski, VT, USA). The infected mice serum antibody was quantified using a gravimetric standard curve (Sigma-Aldrich).

## Histology of gingival tissue

The right mandibles from the polymicrobial-infected and sham-infected (wild type, TLR2$^{-/-}$, and TLR4$^{-/-}$) group's mice ($n = 3$) were decalcified in Immunocal (Decal Chemical, Tallman, NY, USA) for 28 days at 4°C. The decalcified mandibles were embedded in paraffin blocks and sectioned (4 μm) along the mesiodistal plane. Sections were stained with hematoxylin and eosin for histological analysis, and slides were scanned with a ScanScope CS system (Aperio, Vista, CA, USA). The digitally scanned slides were viewed at ×200 magnification objective lens, ScanScope XT (Aperio Technologies, Inc., Vista, CA, USA), and analyzed using the ImageScope program (39, 40). Evidence of inflammation, such as apical junctional epithelial (JE) migration, elongation of rete ridges, resorption of alveolar bone, epithelial hyperplasia, and epithelial edema, was determined (41, 42).

## Measurement of alveolar bone resorption

The impact of polymicrobial infection on alveolar bone was assessed following the histomorphometry procedure (10, 24). After euthanasia, the excised mandibles and maxilla were placed in a beaker and autoclaved to remove the soft tissue. The defleshed bone specimen was cleaned in 3% H$_2$O$_2$ solution and air dried. Two-dimensional imaging of the bone was captured under a 10× stereo dissecting microscope (SteReo Discovery V8, Carl Zeiss Microimaging, Inc., Thornwood, NY, USA). A line tool (AxioVision LE 29A software version 4.6.3) measured horizontal ABR between the cementoenamel junction and alveolar bone crest (12, 42, 43). Three examiners were blinded to measure the ABR of polybacterial and sham infection groups.

## RNA isolation and purification

Mandible from each mouse was homogenized with a handheld rotor-stator homogenizer and sterile individual TissueRuptor disposable probes (Qiagen, Germantown, MD, USA) in the presence of lysis/binding buffer from mirVana miRNA isolation kit (Thermo Fisher Scientific, Waltham, MA, USA; Catalog number: AM1560). Samples were processed following the manufacturer's protocol, and total RNA was extracted. The concentration and purity of the isolated total RNA were determined at 230, 260, and 280 nm in the

TABLE 1   List of 16s rRNA-specific primer sequences for periodontal bacteria

| Bacteria | Forward primer | Reverse primer |
|---|---|---|
| *S. gordonii* | 5′-GTAGCTTGCTACACCATAGA-3′ | 5′-CTCACACCCGTTCTTCTCTT-3′ |
| *F. nucleatum* | 5′-TAAAGCGCGTCTAGGTGGTT-3′ | 5′-ACAGCTTTGCGACTCTCTGT-3′ |
| *P. gingivalis* | 5′-GGTAAGTCAGCGGTGAAACC-3′ | 5′-ACGTCATCCACCCTTCCTC-3′ |
| *T. denticola* | 5′-TAATACCGAATGTGCTCATTTACAT-3′ | 5′-CTGCCATATCTCTATGTCATTGCTCTT-3′ |
| *T. forsythia* | 5′-AAAACAGGGGTTCCGCATGG-3′ | 5′-TTCACCGCGGACTTAACAGC-3′ |

**TABLE 2** Primer sequences of the 15 selected candidate miRNAs[a]

| S. no | miRNA | Mature miR sequence | Selection criteria | |
|---|---|---|---|---|
| | | | Upregulated | Downregulated |
| 1. | mmu-let-7c-5p | UGAGGUAGUAGGUUGUAUGGUU | Poly, *Tf* | Poly, *Pg* |
| 2. | mmu-miR-15a-5p | UAGCAGCACAUAAUGGUUUGUG | Poly, *Pg* | Td, Tf |
| 3. | mmu-miR-22-5p | AGUUCUUCAGUGGCAAGCUUUA | *Pg, Td* | Poly, *Sg* |
| 4. | mmu-miR-30c-5p | UGUAAACAUCCUACACUCUCAGC | *Pg, Td* | *Sg* |
| 5. | mmu-miR-34b-5p | AGGCAGUGUAAUUAGCUGAUUGU | N/A | *Pg, Td, Tf* |
| 6. | mmu-miR-133a-3p | UUUGGUCCCCUUCAACCAGCUG | *Pg, Td* | N/A |
| 7. | mmu-miR-146a-5p | UGAGAACUGAAUUCCAUGGGUU | *Tf* | N/A |
| 8. | mmu-miR-323-3p | CACAUUACACGGUCGACCUCU | *Sg* | Td, Fn, Sg |
| 9. | mmu-miR-339-5p | UCCCUGUCCUCCAGGAGCUCACG | N/A | *Tf, Fn* |
| 10. | mmu-miR-361-5p | UUAUCAGAAUCUCCAGGGGUAC | *Fn, Sg* | Poly, *Sg* |
| 11. | mmu-miR-375-3p | UUUGUUCGUUCGGCUCGCGUGA | Poly, *Sg* | Td, Tf |
| 12. | mmu-miR-423-5p | UGAGGGGCAGAGAGCGAGACUUU | *Tf, Sg* | Td |
| 13. | mmu-miR-720 | AUCUCGCUGGGGCCUCCA | N/A | *Pg, Td, Tf, Fn, Sg* |
| 14. | mmu-miR-132-3p | UAACAGUCUACAGCCAUGGUCG | *Pg, Td* | N/A |
| 15. | mmu-miR-155-5p | UUAAUGCUAAUUGUGAUAGGGGU | N/A | N/A |

[a]Poly, polymicrobial infection (10); *Pg*, *P. gingivalis* monoinfection (11); *Td*, *T. denticola* monoinfection (15); *Tf*, *T. forsythia* monoinfection (13); *Sg*, *S. gordonii* monoinfection (12); *Fn*, *F. nucleatum* monoinfection; and N/A, not available.

NanoDrop Spectrophotometer (ND-1000 Thermo Fisher Scientific). Total RNA was used for RT-qPCR experiments (36).

## Synthesis of cDNA

The purified total RNA was reverse transcribed (20 ng in each sample) into cDNA using miRNA-specific primers and reverse transcription (RT) master mix from TaqMan MicroRNA Reverse Transcription Kit (Thermo Fisher Scientific, USA). The synthesized cDNA is stored at −20°C for further qPCR experiments. The primer sequences for the 15 different miRNAs are shown in Table 2.

## RT-qPCR assay

The reverse transcribed cDNA (2 µL of synthesized cDNA per reaction) was used for qPCR experiments. Reagents used for the qPCR assay were purchased from Life Technologies (Carlsbad, CA, USA). qPCR reactions were performed using the StepOne Real-Time PCR System (Applied Biosystems). Samples were run in duplicates. The protocol includes an initial denaturation step at 95°C for 1 min, followed by 40 cycles consisting of denaturation at 95°C for 10 s, annealing at 50°C for 20 s, and extension at 72°C for 25 s. snoRNA202 was used as an endogenous control for miRNA expression analysis. miRNA levels were normalized to snoRNA202 expression in mouse samples, and cycle threshold (Ct) values, corresponding to the PCR cycle number at which fluorescence emission reaches a threshold above baseline emission, were determined. Relative miRNA expression was calculated using the $2^{-\Delta\Delta Ct}$ method (44, 45).

## Receiver operating characteristic curve analysis

The miRNAs associated with periodontitis (diagnostic miRNAs) were evaluated using the receiver operating characteristics (ROC) curve. RT-qPCR data were analyzed by using the MedCalc Statistical Software version 22.026 (MedCalc Software Ltd, Ostend, Belgium; https://www.medcalc.org), ROC curve analysis was performed, and optimal cut-off values for periodontal miRNAs were determined. The calculated sensitivity and specificity were recorded (46).

## Kyoto Encyclopedia of Genes and Genomes

Kyoto Encyclopedia of Genes and Genomes (KEGG) is an integrated database (47), and its pathways were plotted using the DIANA-miRPath version 3.0 database (48), taking the MIMAT accession number and calculating the false discovery rate (FDR) using the Benjamini and Hochberg method (10).

## Statistical analysis

The statistical significance of ABR data was determined using one-way ANOVA with Dunnett's multiple comparisons set, and analysis was performed using the statistical software Prism 9.4.1 (GraphPad Software, San Diego, CA, USA). The Mann–Whitney $U$ test was used for IgG antibody. For the RT-qPCR data, statistical analysis was performed using the Mann-Whitney $U$ test to determine significant differences in miRNA expression between polymicrobial-infected and sham-infected mice. miR-146a-5p expression between wild-type and TLR2/4 knockout mice was analyzed using ANOVA with *post hoc* Tukey's test. Data are presented as the mean ± standard deviation, and $P < 0.0001$ to $P \leq 0.05$ was considered statistically significant.

## RESULTS

### Gingival colonization of bacteria and host immune response

Analysis of gingival plaque swabs from each mouse (wild type, TLR2$^{-/-}$, and TLR4$^{-/-}$) after bacterial infection showed the presence of bacterial-specific gene amplicons (16s rRNA) in agarose gel electrophoresis. Mice infected with early ecological colonizer *S. gordonii* showed >50% colonization for *S. gordonii* in the first infection cycle (1 IC) and 100% after the second infection cycle (2 IC), indicating that all mice were colonized with gram-positive *S. gordonii*. Gingival swabs from the *F. nucleatum* infection showed >50% colonization after the first IC and reached 100% colonization in the second IC. This confirmed the successful colonization of intermediate periodontal colonizers. Subsequently, 3 weeks after the infection with *P. gingivalis*, *T. denticola,* and *T. forsythia* (2 ICs), mice gingival swabs showed 30%–60% bacterial colonization for all three bacteria and reached 75%–90% colonization after mice received the fourth infection cycle. None of the sham-infected mice were positive at any point for 16s rRNA for any of the five bacteria we analyzed. These results confirmed the successful colonization of five bacteria in the infected mice's oral cavity (Table 3). Serum from the wild-type, TLR2$^{-/-}$, and TLR4$^{-/-}$ mice was evaluated for humoral/IgG immune response against the formalin-killed whole-cell antigens of *F. nucleatum, P. gingivalis, T. denticola,* and *T. forsythia*. Bacteria-specific IgG antibody shown in Fig. 1D was a presentation of IgG fold change in the bacterial infection over the sham infection. The analysis was performed using ordinary two-way ANOVA, and the data are shown as fold change of IgG levels in the infection group over sham group ($n = 16$). A significant increase in IgG immune response was observed in infected mice groups; specifically, the wild-type mice exhibited highly significant serum IgG response (>1,000-fold) to *P. gingivalis* ($P < 0.0001$) and *T. forsythia* ($P < 0.0001$), TLR2$^{-/-}$ and TLR4$^{-/-}$ mice showed significant IgG response to *F. nucleatum* ($P < 0.0001$) and *P. gingivalis* ($P < 0.0001$) (>90-fold), and TLR4$^{-/-}$ mice had a robust level of IgG response to *P. gingivalis* ($P < 0.01$) (>700-fold) (Fig. 1D1–3).

### Deficiency in TLR2 and TLR4 signaling reduces ABR

Polymicrobial infection in wild-type mice resulted in a significant increase in ABR on the mandible lingual ($P < 0.0001$) and maxilla buccal side ($P < 0.01$) than sham-infected mice (Fig. 1B and C1). In contrast, the polymicrobial infection did not induce bone resorption in alveolar bone in the mandible lingual, maxilla palatal, and maxilla buccal sides in TLR2$^{-/-}$ mice, and ABR was not significantly higher in the infected animals than in sham-infected mice (Fig. 1B and C1). Similarly, polymicrobial infection induced no ABR in the mandible lingual, maxilla palatal, and maxilla buccal side in TLR4$^{-/-}$ mice, and

**TABLE 3** Distribution of gingival plaque samples positive for periodontal pathogens' gDNA[a]

| Group/mice/infection | Positive gingival plaque samples (n = 16 mice) | | | | | |
|---|---|---|---|---|---|---|
| | 2 weeks (*Sg*) | 4 weeks (*Sg*) | 6 weeks (*Fn*) | 8 weeks (*Fn*) | 10 weeks (*Pg/Td/Tf*) | 14 weeks *Pg/Td/Tf* |
| Gr I wild-type infection | 11 | 16 | 12 | 16 | 10/9/7 | 15/13/14 |
| Gr II wild-type sham infection | 0 | NC | 0 | NC | 0/0/0 | NC |
| Gr III TLR2$^{-/-}$ infection | 13 | 16 | 11 | 16 | 8/6/9 | 14/12/13 |
| Gr IV TLR2$^{-/-}$ sham infection | 0 | NC | 0 | NC | 0/0/0 | NC |
| Gr V TLR4$^{-/-}$ infection | 10 | 16 | 9 | 16 | 10/7/8 | 14/13/15 |
| Gr VI TLR4$^{-/-}$ sham infection | 0 | NC | 0 | NC | 0/0/0 | NC |

[a]NC, gingival plaque samples not collected.

ABR was not significantly higher in infected mice than the sham-infected mice (Fig. 1B and C1). These findings suggested that resorption of alveolar bone is closely related to gingival inflammation, and mice lacking TLR2 receptors exhibit dampened periodontal inflammatory responses. Polymicrobial infection increased the ABR in the wild-type mice ($P < 0.05$) compared to polymicrobial-infected TLR2$^{-/-}$ mice (Fig. 1C2) and was not significant among the polymicrobial-infected wild-type and polymicrobial-infected TLR4$^{-/-}$ mice (Fig. 1C3), which suggests that TLR4 is less critical than TLR2 in inducing polymicrobial-mediated alveolar bone resorption.

## Deficiency in TLR2 and TLR4 signaling reduces gingival inflammation

Histological examination of the mouse mandible in the infected TLR2$^{-/-}$ and TLR4$^{-/-}$ mice demonstrated minimal apical migration of JE, gingival hyperplasia, and mild inflammatory cellular infiltration in connective tissue (Fig. 1E2 and E3) when compared to sham-infected TLR2$^{-/-}$ and TLR4$^{-/-}$ mice as well as both male and female mice.

These results suggest that TLR2 and TLR4 signaling are critical and play an important role in the inflammatory response against polybacterial infection. In contrast, histological examination of the mouse mandible in the infected wild-type mice showed moderate apical migration of JE, gingival hyperplasia, and inflammatory cellular infiltration in the connective tissue (Fig. 1E1) when compared to sham-infected mice as well as both male and female mice.

## Polymicrobial infection-induced systemic translocation of bacteria to distal organs

To examine whether periodontal bacteria *S. gordonii*, *F. nucleatum, P. gingivalis, T. denticola,* and *T. forsythia* colonized/infected the gingival margins of molar teeth translocated intravascularly to multiple internal organs, we isolated bacterial genomic DNA from the heart, aorta, brain, liver, kidney, and lung, and PCR was performed to detect the presence of specific bacterial gDNA. Bacteria-specific genomic DNA from all five oral microbes (*Sg/Fn/Pg/Td/Tf*) was identified in the heart and lungs, and *P. gingivalis* and *T. denticola* in the brain in wild-type mice (Table 4). Whereas gDNAs of *P. gingivalis* and *T. forsythia* were observed in all organs except the spleen in TLR2$^{-/-}$ mice. The gDNA

**TABLE 4** PCR test results for bacterial dissemination to distal organs[a]

| Group/mice | PCR-positive samples (n = 16) | | | | | |
|---|---|---|---|---|---|---|
| | Heart *Sg/Fn/Pg/Td/Tf* | Lungs *Sg/Fn/Pg/Td/Tf* | Brain *Sg/Fn/Pg/Td/Tf* | Liver *Sg/Fn/Pg/Td/Tf* | Kidney *Sg/Fn/Pg/Td/Tf* | Spleen *Sg/Fn/Pg/Td/Tf* |
| I: wild-type | 13/7/9/11/10 | 8/9/7/2/11 | 0/0/5/3/0 | 0/0/0/0/3 | 8/2/0/0/5 | 2/0/0/0/1 |
| II: sham | 0/0/0/0/0 | 0/0/0/0/0 | 0/0/0/0/0 | 0/0/0/0/0 | 0/0/0/0/0 | 0/0/0/0/0 |
| III: TLR2$^{-/-}$ | 0/0/7/3/12 | 12/0/0/0/16 | 0/0/8/0/8 | 0/0/8/0/7 | 3/0/1/0/10 | 0/0/0/0/0 |
| IV: sham | 0/0/0/0/0 | 0/0/0/0/0 | 0/0/0/0/0 | 0/0/0/0/0 | 0/0/0/0/0 | 0/0/0/0/0 |
| V: TLR4$^{-/-}$ | 0/6/0/0/0 | 0/0/5/0/0 | 0/0/0/0/0 | 0/0/0/0/0 | 0/0/0/0/2 | 0/0/0/0/0 |
| VI: sham | 0/0/0/0/0 | 0/0/0/0/0 | 0/0/0/0/0 | 0/0/0/0/0 | 0/0/0/0/0 | 0/0/0/0/0 |

[a]Group I, wild-type mice/ETSPPI. Group II, wild-type mice/sham infection. Group III, TLR2$^{-/-}$ mice/ETSPPI infection. Group IV, TLR2$^{-/-}$ mice/sham infection. Group V, TLR4$^{-/-}$ mice/ETSPPI infection. Group VI, TLR4$^{-/-}$ mice/sham infection.

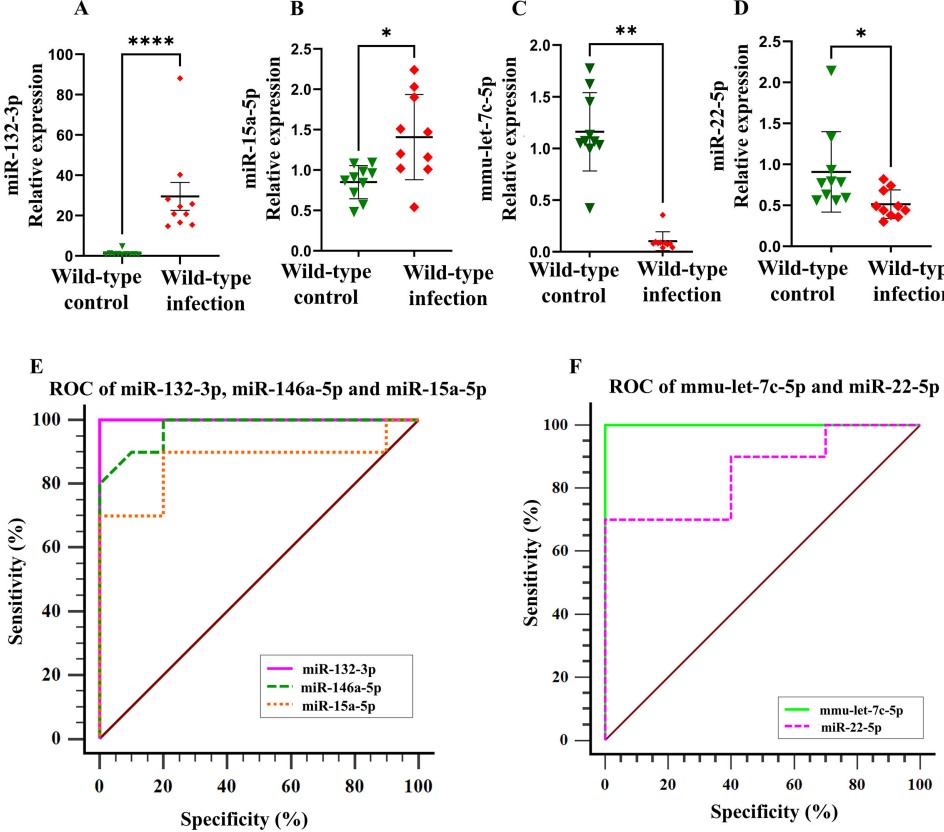

**FIG 2** Relative expression levels were determined by quantitative reverse transcription polymerase chain reaction for selected microRNAs in the mandibles of the polymicrobial-infected, wild-type, and sham-infected mice. (A–D) Relative expression of miR-132-3p, miR-15a-5p, mmu-let-7c-5p, and miR-22-5p, respectively. (E and F) ROC curve of miRNAs that correlate with polybacterial infection-induced periodontitis.

of *S. gordonii* was observed in the lungs and kidneys of infected TLR2$^{-/-}$ mice. The gDNA of *F. nucleatum* was detected in the heart, *P. gingivalis* in the lungs, and *T. forsythia* in the kidney of infected TLR4$^{-/-}$ mice (Table 4).

## miR-146a-5p, miR-15a-5p, and miR-132-3p are elevated, while Let-7c and miR-22 are decreased in wild-type mice

Fifteen miRNA expressions in the mandibles from polymicrobial-infected wild-type and sham-infected mice were analyzed using RT-qPCR. Three miRNAs, miR-132-3p ($P < 0.0001$; Fig. 2A), miR-146a-5p ($P < 0.01$; Fig. 3), and miR-15a-5p ($P < 0.05$; Fig. 2B), were upregulated with an ROC-area under the curve for miR-132-3p (AUC: 1.00), miR-146a-5p (AUC: 0.97), and miR-15a-5p (AUC: 0.87) in the polymicrobial-infected wild-type mice compared to sham-infected mice. miR-132-3p has the highest AUC value, followed by miR-146a-5p and miR-15a-5p. The miRNAs of miR-let-7c-5p (Fig. 2C) and miR-22-5p (Fig. 2D) were downregulated. Figure 2E and F shows the ROC curves for the upregulated and downregulated miRNAs. The remaining 10 miRNA expressions showed no difference between infection and sham infection (Fig. S1; Table 5).

## miR-146a-5p and miR-15a are elevated, while seven miRNAs are downregulated in TLR2$^{-/-}$ mice

The expression profile of 15 miRNAs analyzed in the mandibles of TLR2$^{-/-}$ mice was distinct from wild-type mice. Notably, miR-146a-5p ($P < 0.0001$; Fig. 3) and miR-15a-5p ($P < 0.05$; Fig. 4A) were upregulated with an area under the curve for miR-146a-5p

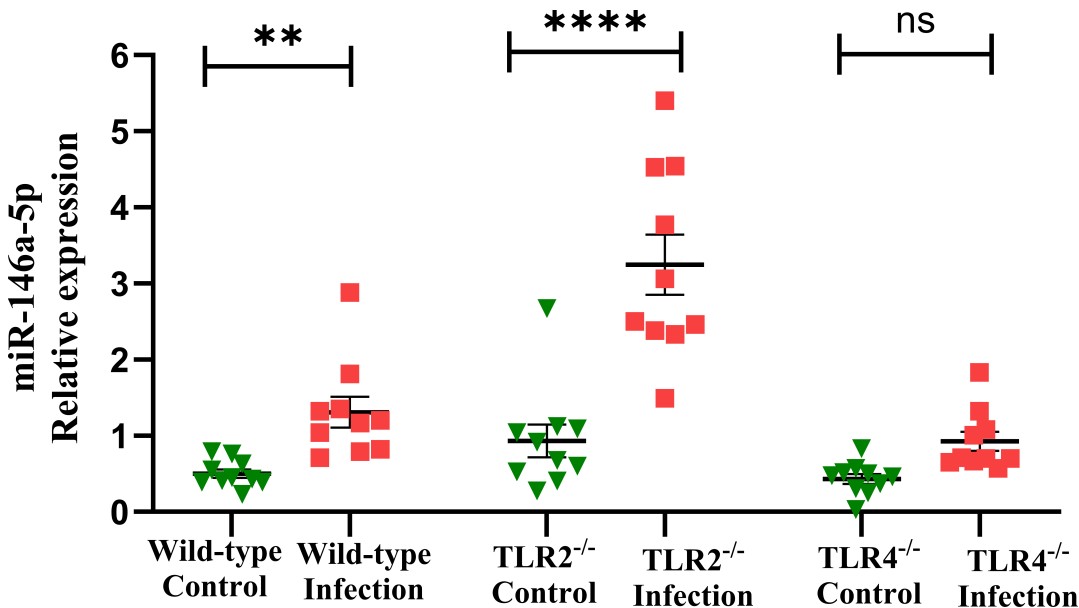

**FIG 3** Relative expression levels were determined for miR-146a-5p by qRT-PCR in the mandibles of polymicrobial-infected wild-type, TLR2[-/-], and TLR4[-/-] mice, as well as their respective sham-infected mice.

(AUC: 0.95) and miR-15a-5p (AUC: 0.82). While both wild-type and TLR2[-/-] mice showed similar upregulation of these two miRNAs, their TLR2[-/-] expression patterns were altered. miR-146a-5p has the highest AUC value compared to miR-15a-5p. The miRNAs of miR-22-5p (Fig. 4B), miR-34b-5p (Fig. 4C), miR-133a-3p (Fig. 4D), miR-339-5p (Fig. 4E), miR-361-5p (Fig. 4F), miR-375-3p (Fig. 4G), and miR-720 (Fig. 4H) were significantly downregulated in TLR2[-/-] mice mandibles. Figure 4J and K shows the ROC curves for the upregulated and downregulated miRNAs. The expression of the remaining miRNA showed no significant variation in infection and sham infection (Fig. S2; Table 5).

## miR-375-3p and other six miRNAs are elevated in the mandibles of TLR4[-/-] mice

The elevated expression levels of miR-146a-5p were significant in the wild-type ($P < 0.01$) and TLR2[-/-] mice ($P < 0.0001$) but not significant in the infected TLR4[-/-] mice (Fig. 3), and the AUC was 0.49 in the ROC analysis. Five of the miRNAs, miR-30c-5p (Fig. 5A), miR-361-5p (Fig. 5B), miR-375-3p (Fig. 5C), miR-720 (Fig. 5D), and mmu-let-7c-5p (Fig. 5E), were upregulated, with a $P$ value $< 0.01$, and two miRNAs, miR-22-5p (Fig. 5F) and miR-323-3p (Fig. 5G), with a $P$ value $< 0.05$. The AUC values for upregulated miRNAs include miR-30c-5p (AUC: 0.88), miR-361-5p (AUC: 0.96), miR-375-3p (AUC: 0.95), miR-720 (AUC: 0.98), mmu-let-7c-5p (AUC: 0.881), miR-22-5p (AUC: 0.8), and miR-323-3p (AUC: 0.85). Figure 5H through J shows the ROC curves for the upregulated and downregulated miRNAs. The remaining seven miRNA expressions showed no difference between infection and sham infection (Fig. S3; Table 5).

## Differentially expressed miRNAs and functional pathway analysis

Predicted functional pathway analysis of the DE miRNAs using KEGG in the polymicrobial-infected wild-type mice mandibles identified several pathways, such as Wnt signaling pathway, focal adhesion pathway, VEGF signaling pathway, B- and T-cell receptor signaling pathway, Axon guidance, TGF-beta signaling pathway, HTLV-1 infection, PI3K-Akt pathway, MAPK signaling pathway, TLR signaling pathway, and osteoclast differentiation pathway (Fig. 6A). Three upregulated miRNAs of wild-type mice have altered the expression of 10 genes in the TLR signaling pathway (Fig. 7A). The two DE upregulated miRNAs from TLR2[-/-]-infected mice mandibles associated with several

TABLE 5  Functions and target genes of miRNAs upregulated by polymicrobial infection

| miRNA | P value | AUC value | Target functions | Target genes[b] |
|---|---|---|---|---|
| **Upregulated miRNAs in the wild-type mice** | | | | |
| miR-146a-5p | 0.01 | 0.97 | Upregulated in the gingiva (49–51) and saliva of periodontitis patients (52). Upregulated in the MM6 cells differentiated in the presence of LPS (TLR4) or zymosan (TLR2) for 96 h (53). | 29 (*PLN, Relb, Cd93, Sgk3, Irak2*) |
| miR-15a-5p | 0.05 | 0.87 | Upregulated in gingival crevicular fluid from periodontitis patients (54) and in serum of myocardial fibrosis patients and serum and myocardial tissues of Ang-II-treated mice (55). Promoting cancer cell invasion and metastasis capabilities through the miR-15a-5p/CXCL17 axis (56). Significantly increased in the exosomes from patients with advanced carotid atherosclerosis (57). Upregulated in the CSF of Alzheimer's disease (58). | 519 (*Tacc1, Bcl2l1, Scn4b, Atg14, Fkrp*) |
| **TLR2<sup>-/-</sup> mice** | | | | |
| miR-146a-5p | 0.01 | 0.95 | Described in the wild-type mice. | |
| miR-15a-5p | 0.05 | 0.82 | Described in the wild-type mice. | |
| **TLR4<sup>-/-</sup> mice** | | | | |
| miR-22-5p | 0.05 | 0.8 | Upregulated in the plasma samples of acute myocardial infarction patients (59, 60). | –[a] |
| miR-30c-5p | 0.01 | 0.88 | Upregulated in gingival crevicular fluid from chronic periodontitis patients (49). Reported in chronic heart failure (61). Upregulated in spared nerve injury rat prelimbic complex (62). Downregulated in atherosclerosis (63). Potential biomarker in the acute rejection after heart transplantation (64). Downregulated in CSF and sera of cryptococcosis patients (65). Upregulated in the urine of post-cardiac surgery patients (66). | 47 (*Egfr, Twf1, Pgr, Six4, Ing1, Reep4*) |
| miR-146a-5p | 0.01 | 0.94 | Described in the wild-type mice. | |
| miR-323-5p | 0.05 | 0.85 | Upregulated in obese periodontitis patients (67). Positive regulator of myogenesis (68), a biomarker for cardiomyopathy (69). Upregulated in the blood samples of coronary heart disease patients and elevated in coronary arteries of coronary heart disease rat model (70). | 25 (*Pira1, Pira2, Tyw1, Atf7, Iars2*) |
| miR-361-5p | 0.01 | 0.96 | Upregulated in acute coronary syndrome (71). Downregulated in *F. nucleatum*-infected colorectal cancer cell lines (HT-29, HCT 116) (72). Upregulated in the serum of acute myocardial injury patients (73). Upregulated in the serum of patients with *Mycobacterium tuberculosis* infection (74). Reported as a biomarker for clinical diagnosis of atherosclerosis (75). | 15 (*Ctbp2, Tfam, Gid4, Ssr1, Eea1*) |
| miR-375-3p | 0.01 | 0.95 | Upregulated in gingival crevicular fluid from chronic periodontitis patients (49), *S. gordonii* induced periodontitis (12). Upregulated in mice exposed to ionizing radiation (76). Upregulated in the hearts of the transverse aortic constriction rat model (77). Negative regulator of osteogenesis (78). Decreased expression was reported in the MPTP-mediated PD mouse model (79). | 19 (*Mtpn, Akap2, Atrn, Rpsa, Yap1*) |
| miR-720 | 0.01 | 0.98 | The most expressed miRNA in chronic and aggressive periodontitis patients (80). Upregulated in HBV-specific CD8+ T cells (81). Upregulated in the serum of hepatitis patients (82). Downregulated in cardiac function for patients after surgery (83). | 9 (*Pon2, Hlcs, Agap1, Ppp6r1*) |
| let-7c-5p | 0.01 | 0.881 | Upregulated in the Bacillus Calmette Guérin-infected macrophage cell lines (84). Inhibits the microglial activation in the cerebral ischemia condition (85). | 21 (*Nf2, Cts8, Lins, Msi2, Myc*) |

[a]"–" indicates that the target gene(s) were not found in the DIANA-miRPath analysis.
[b]The "empty boxes" indicate their explanations were "described in the wildtype mice".

pathways, such as pluripotency of stem cells, TGF-beta signaling pathway, PI3K-Akt pathway, TLR signaling pathway, and different pathways of cancers (Fig. 6B). The two upregulated miRNAs in TLR2<sup>-/-</sup> mice altered nine genes' expression in the TLR signaling pathway (Fig. 7B). Seven downregulated miRNAs of the TLR2 mice have altered 14 gene expressions in the bacterial invasion of the epithelial cell pathway (Fig. 6C). Bacterial invasion of epithelial cells pathway shows the altered gene expression depicted in Fig. 7C. The eight DE upregulated miRNAs from the TLR4<sup>-/-</sup>-infected mice mandibles are associated with the pathways of focal adhesion, MAPK signaling pathway, regulating pluripotent stem cell pathway, platelet activation pathway, TGF-beta signaling pathway, osteoclast differentiation pathway, and ubiquitin-mediated proteolysis pathway (Fig. 6D). The upregulated miRNAs from the TLR4<sup>-/-</sup>-infected mice have altered 27 genes in the osteoclast differentiation pathway.

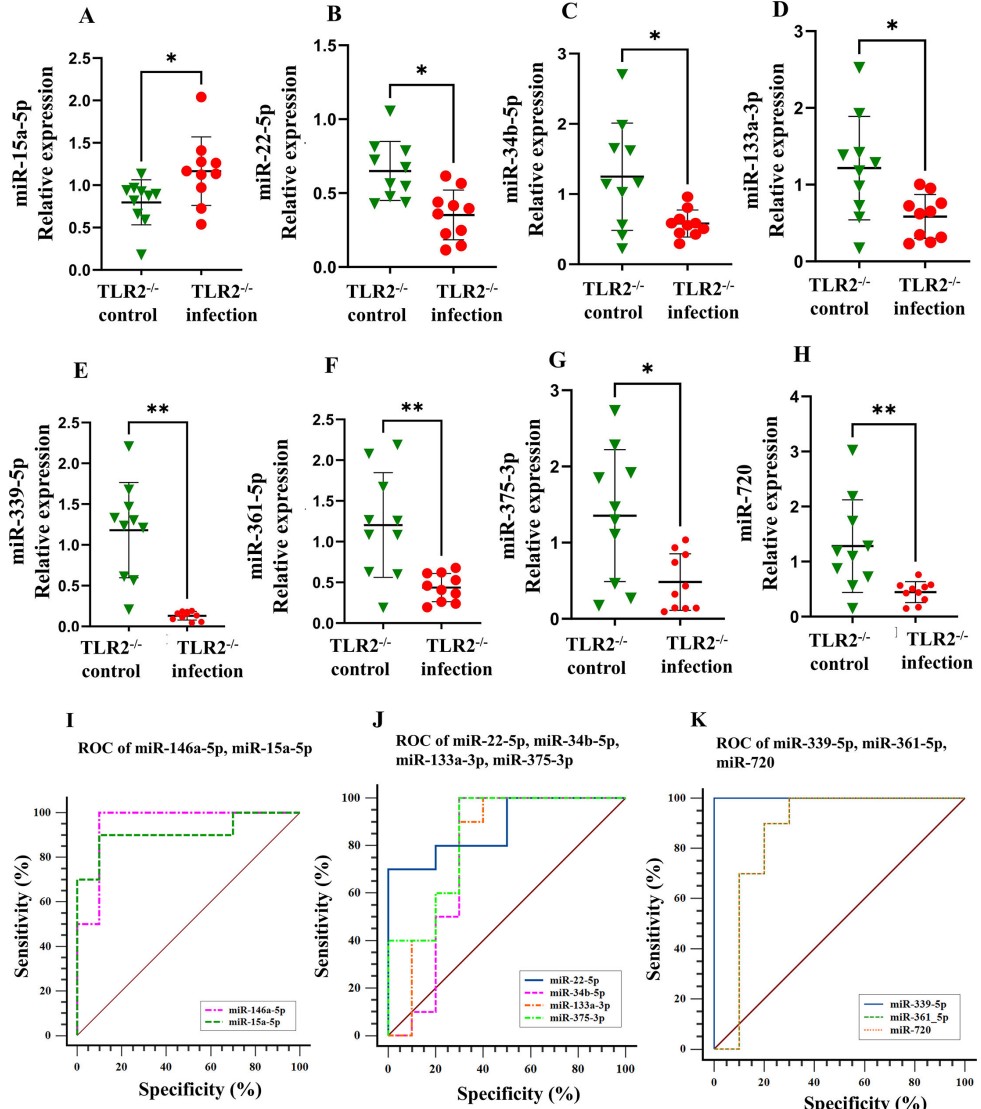

**FIG 4** Relative expression levels were determined by qRT-PCR for selected microRNAs in the mandibles of polymicrobial-infected TLR2[-/-] mice vs TLR2[-/-] sham-infected mice. (A–H) Relative expression of miR-15a-5p, miR-22-5p, miR-34b-5p, miR-133a-5p, miR-339-5p, miR-361-5p, miR-375-3p, and miR-720, respectively. (I–K) ROC curve of miRNAs that correlate with polybacterial infection-induced periodontitis.

## DISCUSSION

The interplay between the dysbiotic microbial community and an abnormal host immune response in the gingival and periodontal tissue contributes to the development of periodontitis (86). The present study broadly investigated the colonization of five periodontal bacteria, infection-induced IgG antibody response, intravascular dissemination, histology, ABR, and miRNA analysis in the wild-type, TLR2[-/-], and TLR4[-/-] mice. Mice infected with ETSPPI showed a significant ABR in wild-type mice but were not observed significantly among the TLR2[-/-] and TLR4[-/-]-infected mice, as in our earlier study (24). Five bacterial gDNAs were detected in the heart and lungs of wild-type mice, while *P. gingivalis* and *T. denticola* were detected in the brain of wild-type mice. Intravascular dissemination of five periodontal bacteria was very limited in TLR2[-/-] and TLR4[-/-] mice. Bacteria accessing blood circulation are commonly observed in the heart valves, liver, and spleen. They can also reach the brain hematogenously. Wild-type mice exhibited significant IgG responses for *P. gingivalis* and *T. forsythia*, while the infected TLR2[-/-]

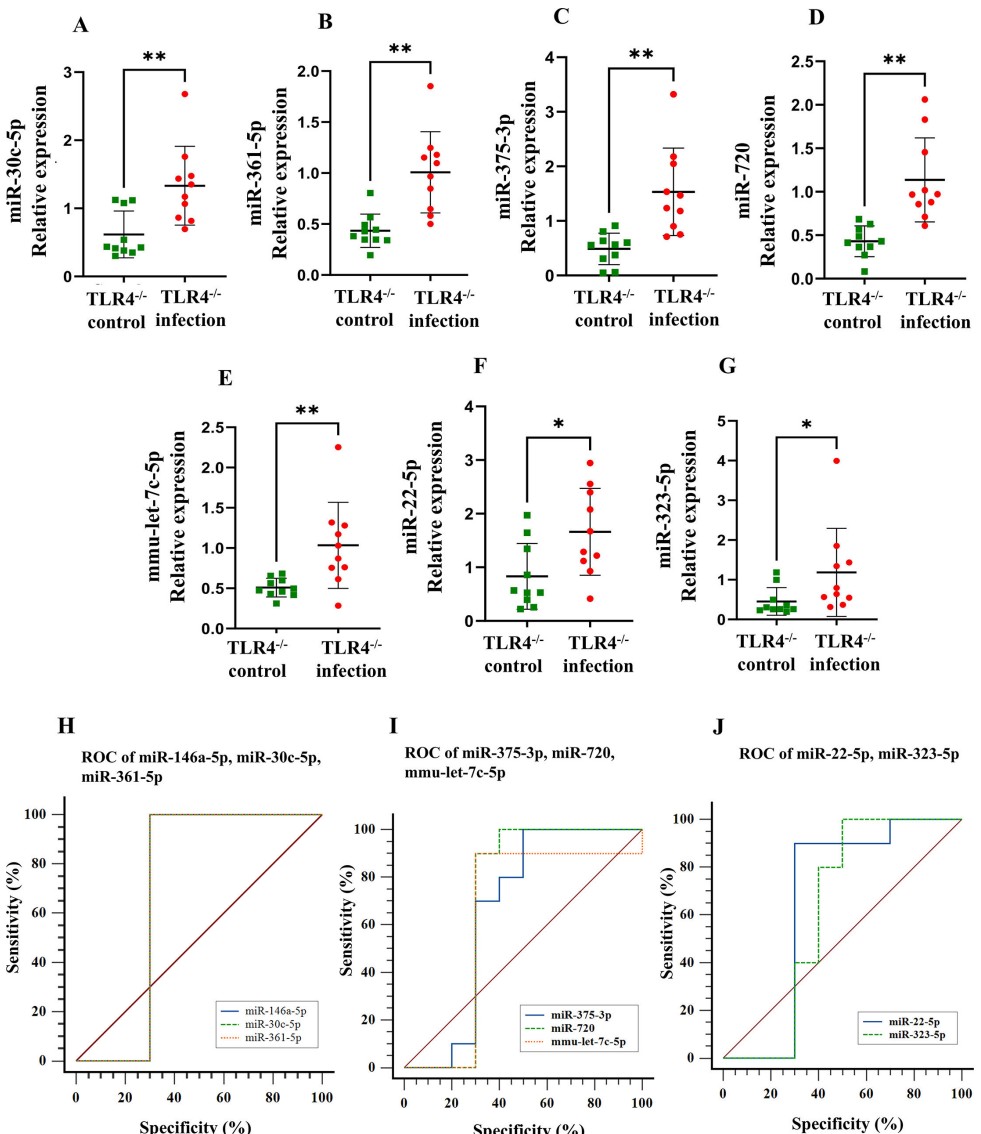

**FIG 5** Relative expression levels were determined by qRT-PCR for selected microRNAs in the mandibles of polymicrobial-infected TLR4$^{-/-}$ mice vs TLR4$^{-/-}$ sham-infected mice. (A–G) Relative expression of miR-30c-5p, miR-361-5p, miR-375-3p, miR-720, mmu-let-7c-5p, miR-22-5p, and miR-323-5p, respectively. (H–J) ROC curve of miRNAs that correlate with polybacterial infection-induced periodontitis.

and TLR4$^{-/-}$ mice showed *P. gingivalis*-specific IgG antibody responses similar to our earlier studies (10, 24). Histological examination of the mouse mandible in the infected TLR2$^{-/-}$ and TLR4$^{-/-}$ mice had minimal apical migration of junctional epithelium, gingival hyperplasia, and mild inflammatory cellular infiltration in connective tissue.

Dysregulated or upregulated miRNAs were identified as potential biomarkers for diagnosing the disease (87, 88). Several miRNAs are used as therapeutic molecules in diseases, such as cancer (89), type-2 diabetes mellitus (90), neurodegenerative diseases (91), and cardiovascular diseases (92). Diagnosing the early stages of PD will prevent disease progression and improve treatment.

We recently reported that periodontal bacteria can induce miRNA expression profiles, which differ from polymicrobial infections and monoinfection (10–13, 15). The selection of miRNAs for RT-qPCR experiments was based on how uniquely they were expressed in six different infections (10–13, 15). For example, mmu-let-7c-5p was expressed in

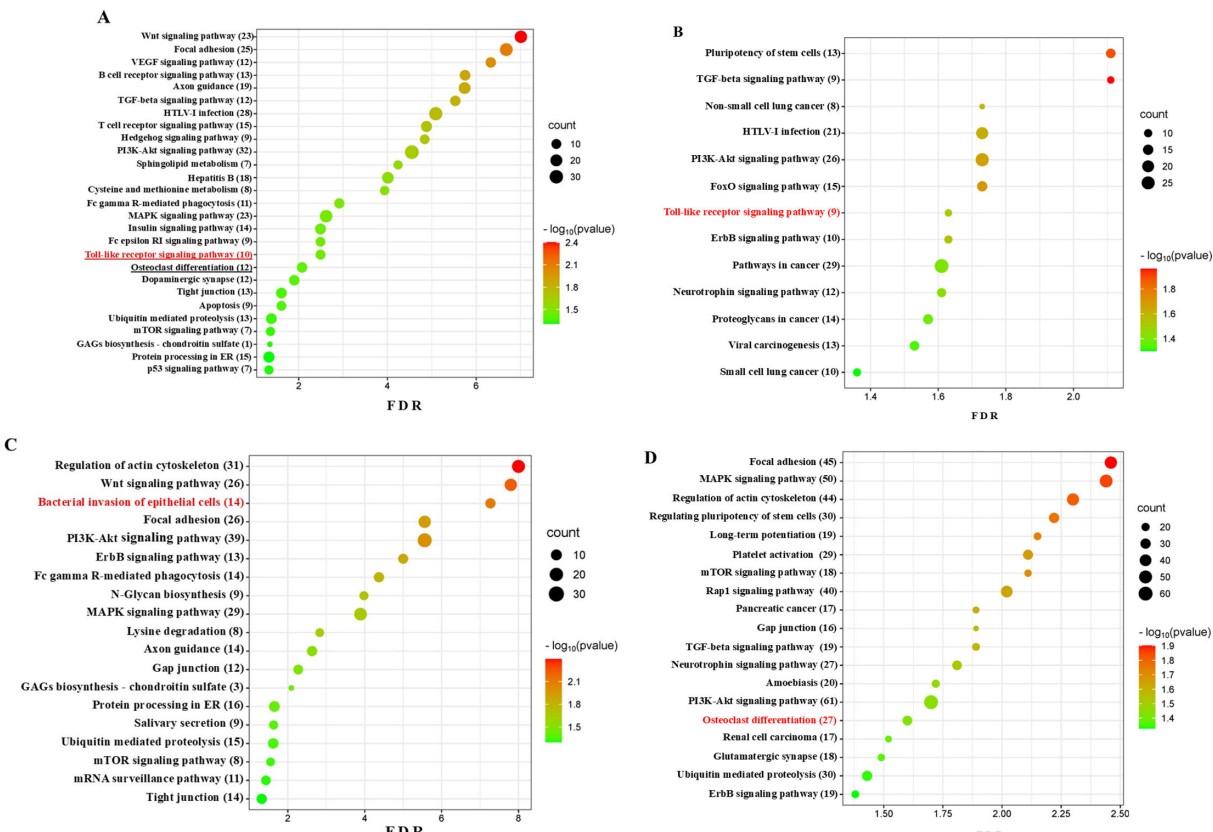

**FIG 6** (A) Bubble plot for upregulated miRNAs in wild-type mice. The predicted functional pathway analysis of DE miRNAs from polymicrobial-infected wild-type mice mandibles. Bubble plot of KEGG analysis on predicted target genes of DE miRNAs in polymicrobial-infected mice compared to sham-infected mice. The KEGG pathways are displayed on the *y*-axis showing the number of genes altered in the pathway in brackets, and the *x*-axis represents the FDR, which means the probability of false positives in all tests. The size and color of the dots represent the number of predicted genes and corresponding *P*-value, respectively. Ten genes were shown to be altered in the Toll-like receptor signaling pathway. (B) Bubble plot for upregulated miRNAs in wild-type and TLR2[-/-] mice. The predicted functional pathway analysis of DE miRNAs from polymicrobial-infected wild-type and TLR2[-/-] mice mandibles. Bubble plot of KEGG analysis on predicted target genes of DE miRNAs in polymicrobial-infected mice compared to sham-infected mice. The KEGG pathways are displayed on the *y*-axis, showing the number of genes altered in the pathway in brackets, and the *x*-axis represents the false discovery rate, which means the probability of false positives in all tests. The size and color of the dots represent the number of predicted genes and corresponding *P*-value, respectively. Nine genes were shown to be altered in the Toll-like receptor signaling pathway. (C) Bubble plot for downregulated miRNAs in TLR2[-/-] mice. The predicted functional pathway analysis of DE miRNAs from polymicrobial-infected TLR2[-/-] mouse mandibles. Bubble plot of KEGG analysis on predicted target genes of DE miRNAs in polymicrobial-infected mice compared to sham-infected mice. The KEGG pathways are displayed on the *y*-axis, showing the number of genes altered in the pathway in brackets, and the *x*-axis represents the FDR, which means the probability of false positives in all tests. The size and color of the dots represent the number of predicted genes and corresponding *P*-value, respectively. (D) Bubble plot for upregulated miRNAs in TLR4[-/-] mice. The predicted functional pathway analysis of DE miRNAs from polymicrobial-infected TLR4[-/-] mouse mandibles. Bubble plot of KEGG analysis on predicted target genes of DE miRNAs in polymicrobial-infected mice compared to sham-infected mice. The KEGG pathways are displayed on the *y*-axis, showing the number of genes altered in the pathway in brackets, and the *x*-axis represents the false discovery rate, which means the probability of false positives in all tests. The size and color of the dots represent the number of predicted genes and corresponding *P*-value, respectively. Twenty-seven genes are known to be altered in the osteoclast differentiation pathway.

polymicrobial infection and monoinfection with *Tf* and *Pg*. miR-146a-5p was expressed in *Tf* monoinfection, miR-15a-5p was expressed in polymicrobial infection and monoinfection with *Pg*, *Td*, and *Tf*. miR-132-5p was expressed in *Pg* and *Td* monoinfection. While miR-22-5p was expressed in polymicrobial infection and monoinfection with *Sg*, *Pg*, and *Td*. miR-323-3p was expressed in monoinfection with *Sg*, *Td*, and *Fn*, and miR-361-5p was expressed in polymicrobial infection and monoinfection with *Sg* and *Fn*. miR-375-3p was expressed in polymicrobial infection and monoinfection with *Td* and *Tf*. miR-720 was expressed in mono-infection with *Sg*, *Fn*, *Pg*, *Td*, and *Tf* (Table 2). Therefore, in this study, we focused on the evaluation of 15 miRNAs, such as let-7c-5p, miR-15a-5p,

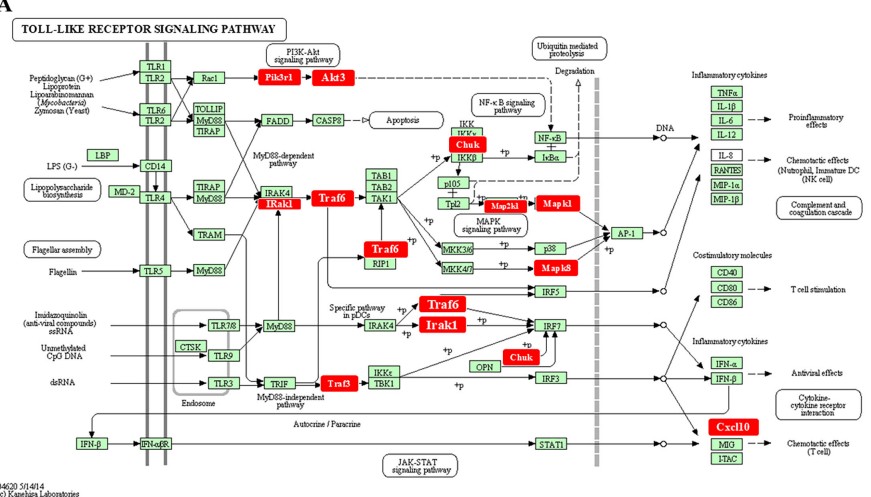

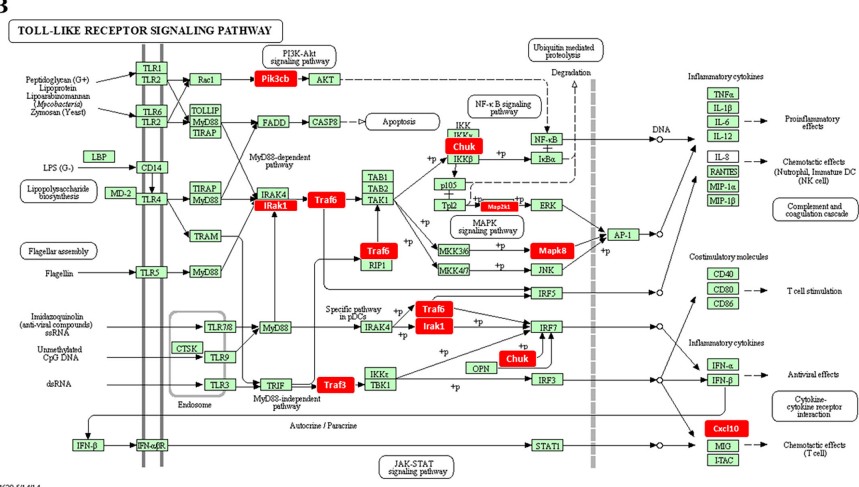

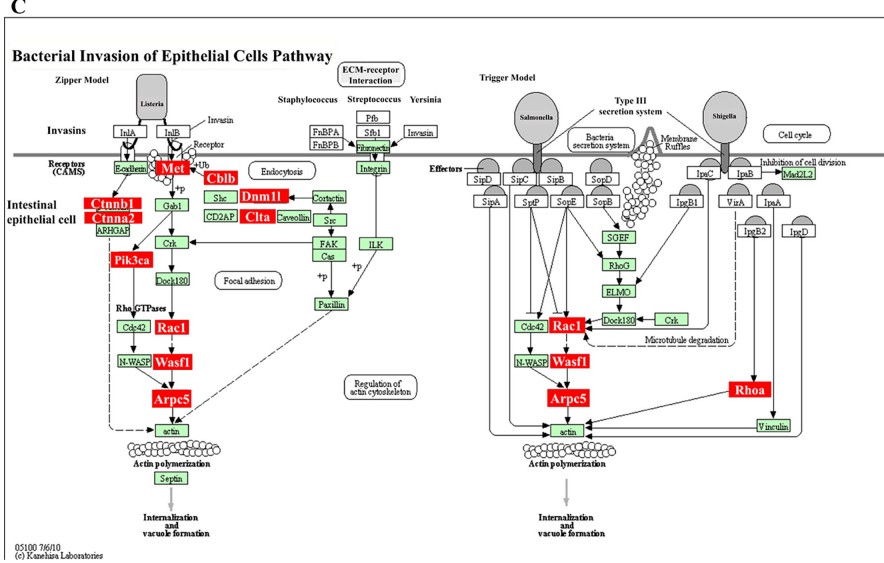

**FIG 7** (A) Upregulated miRNAs in wild-type mice altering 10 genes in Toll-like receptor signaling pathway. The KEGG pathway # mmu04620 was taken from the KEGG website and reused and edited with permission. Significantly differentially expressed genes (identified by KEGG) participate in the TLR signaling pathway. Red boxes indicate significantly increased

Fig 7 (Continued)

expression based on miRNA profiles from NanoString analysis. Green boxes indicate no change in gene expression. Specific families of pattern recognition receptors are responsible for detecting microbial pathogens and generating innate immune responses. Toll-like receptors are membrane-bound receptors identified as homologs of Toll in *Drosophila*. Mammalian TLRs are expressed on innate immune cells, such as macrophages and dendritic cells, and respond to gram-positive or gram-negative bacterial membrane components. Pathogen recognition by TLRs provokes rapid activation of innate immunity by inducing the production of proinflammatory cytokines and upregulation of costimulatory molecules. TLR signaling pathways are separated into two groups: a MyD88-dependent pathway that leads to the production of proinflammatory cytokines with quick activation of NF-$k$B and MAPK, and a MyD88-independent pathway associated with the induction of IFN-beta and IFN-inducible genes, as well as maturation of dendritic cells with slow activation of NF-$k$B and MAPK. (B) Upregulated miRNAs in TLR2$^{-/-}$ mice altering nine genes in the TLR signaling pathway. The KEGG pathway # mmu04620 was taken from the KEGG website and reused and edited with permission. Significantly differentially expressed genes (identified by KEGG) participate in the Toll-like receptor signaling pathway. Red boxes indicate significantly increased expression based on miRNA profiles from NanoString analysis. Green boxes indicate no change in gene expression. Specific families of pattern recognition receptors are responsible for detecting microbial pathogens and generating innate immune responses. TLRs are membrane-bound receptors identified as homologs of Toll in *Drosophila*. Mammalian TLRs are expressed on innate immune cells, such as macrophages and dendritic cells, and respond to the membrane components of gram-positive or gram-negative bacteria. Pathogen recognition by TLRs provokes rapid activation of innate immunity by inducing the production of proinflammatory cytokines and upregulation of costimulatory molecules. TLR signaling pathways are separated into two groups: a MyD88-dependent pathway that leads to the production of proinflammatory cytokines with quick activation of NF-kB and MAPK, and a MyD88-independent pathway associated with the induction of IFN-beta and IFN-inducible genes, as well as maturation of dendritic cells with slow activation of NF-kB and MAPK. (C) Downregulated miRNAs in TLR2$^{-/-}$ mice altering 14 genes in the bacterial invasion of epithelial cell pathway. The KEGG pathway # mmu04620 was taken from the KEGG website and reused and edited with permission. Significantly, DE genes (identified by KEGG) participate in the TLR signaling pathway. Red boxes indicate significantly increased expression based on miRNA profiles from NanoString analysis. Green boxes indicate no change in gene expression. Many pathogenic bacteria can invade phagocytic and non-phagocytic cells and colonize them intracellularly, then become disseminated to other cells. Invasive bacteria induce their own uptake by non-phagocytic host cells (e.g., epithelial cells) using two mechanisms referred to as the zipper model and trigger model. *Listeria, Staphylococcus, Streptococcus*, and *Yersinia* are examples of bacteria that enter using the zipper model. These bacteria express proteins on their surfaces that interact with cellular receptors, initiating signaling cascades that result in close apposition of the cellular membrane around the entering bacteria. *Shigella* and *Salmonella* are examples of bacteria entering cells using the trigger model. These bacteria use type III secretion systems to inject protein effectors that interact with the actin cytoskeleton.

miR-22-5p, miR-30c-5p, miR-34b-5p, miR-133a-3p, miR-146a-5p, miR-323-3p, miR-339-5p, miR-375-3p, miR-361-5p, miR-423-5p, miR-720, miR-155-5p, and miR-132-3p, which revealed the expression difference between polymicrobial infection-treated mandibles and the healthy mice mandibles in the cohorts of wild-type, TLR2$^{-/-}$, and TLR4$^{-/-}$ mice. The miRNA analysis in this study provided PD-specific miRNA expression patterns in the wild-type mice and the miRNA hallmarks for periodontitis in the TLR-deficient mice.

This study found that miR-146a-5p, miR-15a-5p, and miR-132-3p were upregulated in wild-type infected mice. miR-146a-5p and miR-15a-5p were upregulated in TLR2$^{-/-}$-infected mice. Seven miRNAs (miR-30c-5p, miR-22-5p, miR-323-3p, miR-361-5p, miR-375-3p, miR-720, and let-7c-5p) were upregulated in the TLR4$^{-/-}$-infected mice. There is clear evidence that TLR2 and TLR4 mediate the early inflammatory response (93) or pro-inflammatory cytokine production (94, 95) and induced inflammation in PD development (24). Notably, miR-146a-5p was upregulated uniquely among the wild-type and TLR2$^{-/-}$ infection groups. miR-146a is a master regulator involved in controlling multiple TLR signaling pathways, including TLR4/2, but also other TLRs (96, 97). miR-146a-5p was reported as an inflammation-induced miRNA in PD (49–51).

Several studies reported that miRNAs can intervene in the initiation and modulate the expression of TLRs and multiple components of TLR-signaling pathways, such as signaling proteins, transcription factors, and cytokines (98, 99). Activated TLRs induce several TLR-responsive upregulated and downregulated miRNAs (99). The upregulated miRNA predicts base pairing with sequence in the 3′ UTR of the responsive protein genes

and inhibits essential protein expression for TLR activation and affects TLR signaling (100). Activated TLR2 and TLR4 receptors increased the expression of miR-155, miR-146, miR-147, and miR-9 (99). Activated TLR4 alone upregulated miR-21, miR-223, miR-125b, let-7e, and miR-27b, and downregulated miR-125b, let-7i, and miR-98. miR-146 has verified TLR-signaling targets of IRAK-1, IRAK-2, and TRAF6 in the TLR signaling pathways (99). miR-155 has verified TLR-signaling targets of MYD88, TAB2, and IKKε, and transcription factor targets of FOXP3 and C/EBPβ, cytokine targets of TNF, and regulators of SHIP1 and SOCS1 (99).

Based on the above evidence, we interpret that the DE (upregulated and downregulated) miRNAs in wild-type, TLR2$^{-/-}$, and TLR4$^{-/-}$ mice could be responsive miRNAs for the activation of TLR2, TLR4, or both. miR-146a-5p, commonly upregulated in wild-type, TLR2$^{-/-}$, and TLR4$^{-/-}$ mice, could be a target for TLR signaling in both TLR2/4 receptors. The upregulated miR-15a-5p could target the TLR pathway through TLR4 receptor activation. The ROC curve analysis revealed miR-146a-5p as the most predictable biomarker in PD in the infected mice. The AUC value for 146a-5p was 0.975 in the wild-type mice and 0.950 and 0.940 in the TLR2$^{-/-}$ and TLR4$^{-/-}$ mice, respectively. This positive correlation confirms miR-146a-5p as a polymicrobial-induced PD-inflammatory biomarker among the wild-type, TLR2$^{-/-}$, and TLR4$^{-/-}$-infected mice. Earlier reports suggest that miR-15a-5p modulates the natural killer and CD8$^+$ T-cell activation (101) and is a vital regulator in inflammation-induced sepsis (102). miR-15a-5p was reported as an upregulated miRNA in the gingival crevicular fluid of the PD patients (54), and it was also upregulated in the wild-type (AUC:0.870) and TLR2$^{-/-}$ mice (AUC:0.820) with the correlations in AUC values. The present study confirmed that miR-146a-5p and miR-15a-5p are unique TLR2$^{-/-}$-independent PD markers and recommended molecules of periodontal therapy. However, it is acknowledged that, in human PD, the exact combination of oral bacteria involved in disease pathogenesis remains unclear; further work is needed to examine the role of these miRNAs using larger scale studies in humans.

In the differentially expressed study, eight miRNAs (miR-22-5p, miR-323-5p, miR-30c-5p, mmu-let-7c-5p, miR-146a-5p, miR-375-3p, miR-361-5p, and miR-720) were upregulated in the mandibles of TLR4$^{-/-}$-infected mice. The DE upregulated miRNAs (miR-30c-5p, miR-375-3p, and miR-720) are associated with PD (49, 80). Several of the upregulated miRNAs were associated with systemic diseases, such as acute myocardial infarction (miR-22-5p) (59, 60), coronary heart disease (miR-323-3p) (70), and acute coronary syndrome (miR-361-5p) (71). The studies from Wang et al. (103) also reported that miR-361-5p is an upregulated miRNA with ROC-AUC:0.891 in carotid artery stenosis patients. ROC analysis in the present study confirmed a positive correlation for miR-361-5p (AUC:0.960), miR-375-3p (AUC:0.950), and miR-miR-720 (AUC:0.980) as the most predictive markers for mice infected with periodontitis. miR-361-5p serves as a biomarker to predict acute coronary syndrome (71). The AUC values for miR-30c-5p and mmu-let-7c-5p were identical (AUC:0.880), and miR-22-5p (AUC:0.800) and miR-323-3p (AUC:0.850) had the lowest AUC values. miR-22-5p was consistently downregulated in wild-type and TLR2$^{-/-}$ mice, suggesting that its regulation may occur through TLR4-mediated signaling. miR-22-5p, miR-361-5p, miR-375-3p, and miR-720 downregulated in TLR2$^{-/-}$ mice were upregulated in TLR4$^{-/-}$ mice. These miRNAs could be TLR-independent signaling markers/ligands. Let-7c-5p in wild-type (downregulated) and TLR4$^{-/-}$ (upregulated) mice could be a TLR target marker/ligand-activated through TLR2 receptors.

Using the DIANA-miRPath web tool and KEGG analysis, we found that upregulated and downregulated DE miRNAs altered the gene expression in specified pathways further to inflammation, infection, and PD. The KEGG-predicted functional pathway analysis revealed significant findings among the wild-type, TLR2$^{-/-}$, and TLR4$^{-/-}$ mice. The miRNAs upregulated in wild-type, TLR2$^{-/-}$, and TLR4$^{-/-}$ mice have the target pathways of 49, 51, and 67, respectively. The miRNAs upregulated in wild-type and TLR2$^{-/-}$ mice or wild-type and TLR4$^{-/-}$ mice have 43 similar target pathways. The upregulated miRNAs in TLR2$^{-/-}$ and TLR4$^{-/-}$ mice have 37 similar target pathways.

The upregulated and downregulated miRNAs are the keys to fine-tune controls of gene expression for many of the regulatory proteins associated with the presented KEGG pathways. The pathways unique to wild-type and TLR2$^{-/-}$ mice, as well as the comparison study between wild-type and TLR4$^{-/-}$ mice, are as follows: Epstein-Barr virus infection, measles, natural killer cell-mediated cytotoxicity, NF-kappa B signaling pathway, tyrosine metabolism, viral carcinogenesis in TLR2 mice, and adherens junction, ECM-receptor interaction, gap junction, salivary secretion, vascular smooth muscle contraction, etc., among the 30 pathways in TLR4$^{-/-}$ mice. This miRNA expression kinetics was used to identify target genes that can influence the function of PD-related biological pathways. Using the DIANA-miRPath web tool and KEGG analysis, we found that the DE-miRNAs, both upregulated and downregulated, altered the gene expression in specified pathways further toward inflammation, infection, and PD. The results demonstrated that polymicrobial infection induced PD in wild-type mice with significant ABR and elevated the expression of miR-146a-5p, miR-132-3p, and miR-15a-5p. Polymicrobial infection-induced PD in TLR2$^{-/-}$ and TLR4$^{-/-}$ mice, but none showed significant ABR compared to their sham infection. miR-146a-5p and miR-15a-5p were upregulated in the TLR2$^{-/-}$ mice, whereas miR-146a-5p and the other seven miRNAs were upregulated in TLR4$^{-/-}$ mice. miR-146a-5p and miR-15a-5p could be markers for PD in wild-type and TLR2$^{-/-}$ mice. miR-146a-5p, miR-30c-5p, miR-323-3p, and miR-375-3p, which were upregulated in TLR4$^{-/-}$ mice, have also been reported in human PD and are therefore considered potential markers of PD in TLR4$^{-/-}$ mice.

We hypothesize that during severe periodontitis, subgingival periodontal bacteria might enter directly into the bloodstream through the subgingival ulcerated gingival epithelium, causing bacterial translocation to distant organs, and colonize in multiple organs (104). Future studies will aim to investigate the mechanisms of oral bacterial translocation to multiple systemic organs (heart, lungs, liver, spleen, kidney, and brain) and its impact on systemic inflammation. *T. denticola* virulence factors activate TLR2 (105), and *Staphylococcus aureus* lipoprotein stimulates NF-κB pathway (106). Interestingly, bone marrow-derived macrophages respond to *P. gingivalis* through activation and nuclear translocation of interferon regulatory factor 3 (107). Human monocytes infected with *P. gingivalis* gingipains induce c-Jun/c-Fos (AP-1) and phosphorylated NF-*k*B (108), as well as signal transducer and activator of transcription (STAT 1/STAT3). Quantifying these key transcription factors and exploring the alternative PRRs that may compensate for TLR deficiencies could have potential implications in elucidating the molecular components involved in TLR activation. We acknowledge that the results from our current study are based on an excellent murine model of PD. However, in order for this work to have an impact on human disease, it is logical to plan detailed studies with human disease.

In conclusion, this study explored the expression levels of selected 15 miRNAs that were significantly altered (upregulated/downregulated) during polymicrobial and monobacterial infections in C57BL6/J wild-type, TLR2$^{-/-}$, and TLR4$^{-/-}$ mice, compared to sham-infected controls using RT-qPCR. The data established that miR-146a-5p, miR-15a-5p, miR-30c-5p, miR-361-5p, miR-375-3p, miR-720, and mmu-let-7c-5p possibly function as beneficial biomarkers for the therapeutic targeting of periodontitis. In addition, these data indicate that the above miRNAs may play a critical role (markers/ligands) in the induction of gingival inflammation by regulating TLR2/4 signaling pathways. The increase in the level of certain miRNAs, for example, miR-146a, in all these different strains may implicate that these miRNAs can be important biomarkers independent of whether the PD pathogenesis focuses more heavily on TLR4 or TLR2 —depending in part on the specific oral pathogen(s) involved. As discussed, these interesting data, however, cannot be directly interpreted for the human disease until large-scale clinical studies are designed to examine whether miRNA biomarkers can predict disease progression or sensitivity to treatment.

## ACKNOWLEDGMENTS

This work was supported by NIH National Institute of Dental and Craniofacial Research (Grant no. R01 DE028536) to L.K. and E.K.L.C.

S.J. performed mouse experiments and analyzed the data. A.Y. and J.O. did the molecular analysis of distal organ dissemination; P.G. performed colony PCR for oral swabs; A.R.L.M., E.C., S.R.N., and T.D. performed the alveolar bone resorption study; J.W. performed data analysis for RT-PCR. I.B. was responsible for histology interpretations. L.K. was responsible for the conception, experimental design, analysis, initial drafting, editing, supervision, project administration, interpretation, and funding acquisition. L.K. and S.J. participated in the final revision and editing of the manuscript. L.K. and E.K.L.C. are the Principal Investigators of the NIH (NIDCR) study. All authors have read and agreed to the published version of the manuscript.

The authors declare no conflict of interest. The funders had no role in the design of the study; in the collection, analyses, or interpretation of data; in the writing of the manuscript; or in the decision to publish the results.

## AUTHOR AFFILIATIONS

[1]Department of Periodontology, College of Dentistry, University of Florida, Gainesville, Florida, USA

[2]Department of Oral Biology, College of Dentistry, University of Florida, Gainesville, Florida, USA

[3]Department of Oral and Maxillofacial Diagnostic Sciences, College of Dentistry, University of Florida, Gainesville, Florida, USA

## AUTHOR ORCIDs

Syam Jeepipalli http://orcid.org/0000-0002-2769-8619
L. Kesavalu http://orcid.org/0000-0002-8828-8656

## FUNDING

| Funder | Grant(s) | Author(s) |
| --- | --- | --- |
| National Institute of Dental and Craniofacial Research | R01 DE028536 | L. Kesavalu |
| | | Edward K. L. Chan |

## AUTHOR CONTRIBUTIONS

Syam Jeepipalli, Conceptualization, Data curation, Formal analysis, Funding acquisition, Investigation, Methodology, Project administration, Resources, Software, Supervision, Validation, Visualization, Writing – original draft, Writing – review and editing | Parvathi Gurusamy, Data curation, Formal analysis, Investigation, Methodology, Software, Supervision, Validation, Visualization, Writing – original draft, Writing – review and editing | Ana Rafaela Luz Martins, Data curation, Formal analysis, Investigation, Methodology, Software, Supervision, Validation, Visualization, Writing – review and editing | Eduardo Colella, Data curation, Formal analysis, Investigation, Methodology, Software, Supervision | Sandhya R. Nadakuditi, Data curation, Formal analysis, Investigation, Methodology, Software, Supervision | Tushar Desaraju, Data curation, Formal analysis, Investigation, Methodology, Software, Supervision | Ashitha Yada, Data curation, Formal analysis, Investigation, Methodology, Software, Validation | Jennifer Onime, Data curation, Formal analysis, Investigation, Methodology, Validation | John T. William, Formal analysis, Investigation, Methodology, Software, Validation | Indraneel Bhattacharyya, Formal analysis, Investigation, Methodology, Software, Validation | Edward K. L. Chan, Conceptualization, Formal analysis, Funding acquisition, Investigation, Methodology, Project administration, Software, Supervision, Validation, Writing – review and editing | L. Kesavalu, Conceptualization, Formal analysis, Funding acquisition, Investigation,

Methodology, Project administration, Resources, Supervision, Validation, Visualization, Writing – original draft, Writing – review and editing

## ETHICS APPROVAL

All animal procedures were approved by the University of Florida Institutional Animal Care and Use Committee (IACUC) under protocol number 202200000223.

## ADDITIONAL FILES

The following material is available online.

### Supplemental Material

**Supplemental figures (Spectrum00160-25-s0001.docx).** Fig. S1 to S3.

### Open Peer Review

**PEER REVIEW HISTORY An accounting of the reviewer comments and feedback. (review-history.pdf).**

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
