## [Reviewer comments · Microbiology Spectrum]

Microbiology Spectrum

Altered microRNA Expression Correlates with Reduced TLR2/4-Dependent Periodontal Inflammation and Bone Resorption Induced by Polymicrobial Infection

Syam Jeepipalli, Parvathi Gurusamy, Ana Martins, Eduardo Colella, Sandhya Nadakuditi, Tushar Desaraju, Ashitha Yada, Jennifer Onime, John William, Indraneel Bhattacharyya, Edward K.L. Chan, and Kesavalu Lakshmyya

Corresponding Author(s): Kesavalu Lakshmyya, University of Florida

Review Timeline:

Submission Date:	January 15, 2025
Editorial Decision:	March 5, 2025
Revision Received:	April 16, 2025
Editorial Decision:	June 12, 2025
Revision Received:	June 26, 2025
Accepted:	July 14, 2025

Editor: Cassio Almeida-da-Silva

Reviewer(s): Disclosure of reviewer identity is with reference to reviewer comments included in decision letter(s). The following individuals involved in review of your submission have agreed to reveal their identity: Kamoru A ADEDOKUN (Reviewer #2)

Transaction Report:

DOI: <https://doi.org/10.1128/spectrum.00160-25>

Re: Spectrum00160-25 (Altered microRNA Expression Correlates with Reduced TLR2/4-Dependent Periodontal Inflammation and Bone Resorption Induced by Polymicrobial Infection)

Dear Dr. Kesavalu Lakshmyya:

Thank you for the privilege of reviewing your work. Below you will find my comments, instructions from the Spectrum editorial office, and the reviewer comments.

Revision Guidelines

- Upload point-by-point responses to the issues raised by the reviewers in a file named "Response to Reviewers," NOT in your cover letter. In your answer, please include page and lines in which the modifications were made in the manuscript.
- Upload a compare copy of the manuscript (without figures) as a "Marked-Up Manuscript" file.
- Upload a clean .DOC/.DOCX version of the revised manuscript and remove the previous version.
- Each figure must be uploaded as a separate, editable, high-resolution file (TIFF or EPS preferred), and any multipanel figures must be assembled into one file.
- Any supplemental material intended for posting by ASM should be uploaded with their legends separate from the main manuscript. You can combine all supplemental material into one file (preferred) or split it into a maximum of 10 files with all associated legends included.

Sincerely,
Cassio Almeida-da-Silva
Editor
Microbiology Spectrum

Reviewer #1 (Comments for the Author):

The manuscript describes the study of miRNA expressions that are correlated with TLR2/4 that are induced by various oral bacterial treatment. The study includes a large volume of mouse samples and covers a variety of experiments. The major drawback of this manuscript is the data presentation and labelling which making the readers hard to follow. The authors have to address them before publication.

Major comments:

>>Figure 1C. What do 1,2,3,4,5 mean? although it seems to be mentioned in line 286-291, but 1C4 and 1C4 seems not described. The authors should at least label them clearly in the figure. To compare WT and TLR2/TLR4, the authors should include all the following groups on a single graph: WT control, WT infection, TLR2 control, TLR2 infection, TLR4 control, and TLR4 infection. A two-way ANOVA statistical test should be performed to assess these comparisons. It is noted that the TLR2 and TLR4 groups show an increased ABR compared to the WT control group in the uninfected conditions. The authors need to explain if this increase is expected.

>>Fig 1D. It is unclear why error bars and statistical tests are not present.

>>Fig 2/3/4. Why the same miRNA can not be compared in one figure? For example: miR-146a-5p were tested in all 6 mouse samples (WT/TLR2-/TLR4-; control/infection), and all the infection samples are elevated in the miR-146a-5p.

>>why ROC/AUC value are used in Fig 2/3/4 and table 5, ROC/AUC is commonly used when comparing groups in medicine/pharmacokinetic, while it is less common to compare expression in biological samples. As the statistical comparison (P-value) has been shown already, the ROC /ACU seems not providing additional information and is also not easy to be understood by general biological readers that the authors could consider to remove them.

>>Figure 5A-C are not easy to understand, could they be combined in one figure/alternative way to show the differences between groups?

Reviewer #2 (Comments for the Author):

General Assessment

I acknowledge the authors on their efforts investigating the role of Toll-like receptor (TLR) signaling in regulating microRNA (miRNA) expression in periodontitis using wild-type, TLR2^{-/-}, and TLR4^{-/-} mouse models. The study provides valuable insights into the relationship between TLR2/4 signaling and periodontal inflammation, alveolar bone resorption (ABR), and systemic bacterial dissemination. However, several key aspects require further clarification and additional analysis to strengthen the manuscript's scientific impact. Below are detailed comments and suggestions for improvement.

Major Concerns

1. Expression of miR-146a-5p in Wild-Type and Knockout Mice:

Concern: The study reports upregulation of miR-146a-5p in both wild-type and TLR2^{-/-}/TLR4^{-/-} mice but does not explore differential expression or specific roles within each genotype.

Recommendation: I would suggest that the authors conduct pairwise comparisons of miR-146a-5p expression between wild-type and knockout groups using statistical tests such as ANOVA with post hoc Tukey's tests. This will determine whether its expression is quantitatively altered due to TLR2/4 signaling loss. Additionally, investigate whether other pattern recognition receptors (e.g., NOD-like receptors) contribute to miR-146a-5p upregulation in knockout mice.

2. Type of Knockout Model and Cell Types Implicated:

Concern: The manuscript does not specify whether the TLR2^{-/-} and TLR4^{-/-} mice are global or conditional knockouts. This distinction is critical because TLRs are expressed in diverse cell types, including gingival epithelial cells, fibroblasts, macrophages, neutrophils, and osteoclasts.

Recommendation: The authors should clarify whether the knockout models are global or conditional and specify the targeted cell types. They need to perform cell-type-specific RT-qPCR or immunohistochemistry (IHC) to determine which cells are primarily responsible for miRNA expression. Flow cytometry can be used to quantify TLR2/4 expression in immune cells and gingival tissues.

3. Pathways and Transcription Factors (TFs) Implicated:

Concern: The manuscript mentions the TLR signaling pathway but does not provide detailed insights into the transcription factors (TFs) connecting TLR activation to miRNA expression.

Recommendation: I would recommend that the authors perform Western blot or RT-qPCR to quantify key TFs such as NF- κ B (p65), IRF3, AP-1, c-Jun, STAT1, and STAT3. Investigate whether miR-146a-5p is regulated via NF- κ B by measuring phosphorylated NF- κ B levels and using inhibitors to assess its effect on miRNA expression. Explore alternative PRRs that may compensate for TLR deficiencies.

Minor Concerns

4. Validation of miRNA Biomarkers in Human Samples:

While the mouse model is appropriate for mechanistic insights, validation in human PD patients is essential to confirm the clinical relevance of miRNAs like miR-146a-5p and miR-15a-5p.

Suggestion: To improve the rigor of this study, the investigators need to conduct miRNA profiling using gingival crevicular fluid (GCF) or saliva samples from PD patients and healthy controls.

5. Long-Term Effects of Chronic Infection:

There is a limited time frame in this analysis. The study focuses on acute infection (10 infection cycles).

Suggestion: The long-term effects of chronic infection should be evaluated to understand the progression of PD and sustained miRNA expression changes. I would suggest extending the analysis beyond acute infection to evaluate sustained miRNA expression changes during chronic periodontitis.

6. Systemic Bacterial Dissemination:

How the bacteria disseminated was not totally skipped.

Suggestion: It is inquisitive to know how the organisms traffic systemically, especially blood-brain barrier permeability and immune evasion mechanisms that facilitate bacterial dissemination. Thus, I would suggest that the investigators should further investigate the mechanisms of bacterial translocation to systemic organs and its impact on systemic inflammation.

7. Histological Analysis:

It is disappointing to see that the authors only used basic stain tech to answer questions that have to do with various markers and proteins in tissue. The qualitative H&E staining provides limited information on inflammatory cell types.

Suggestion: The authors should use stains such as IHC to identify inflammatory cell types (macrophages, neutrophils, T cells) and quantify cytokine expression (TNF- α , IL-1 β , IL-6) in gingival tissues.

Overall Manuscript Organization and Clarity

Abstract: Pls condense the key findings and emphasize the clinical relevance of miRNA biomarkers.

Introduction: I would suggest an improvement to enhance the discussion of miRNAs' clinical significance and their potential as diagnostic tools.

Methods: Pls, clearly state the rationale for selecting the 15 miRNAs and describe the knockout models in detail.

Results: I recommend the authors should simplify figures for clarity, combining ROC curves into a single panel.

Discussion: This section is highly deficient. I recommend that the investigators provide a more comprehensive interpretation of the molecular pathways and potential therapeutic applications of identified miRNAs.

General Assessment

I acknowledge the authors on their efforts investigating the role of Toll-like receptor (TLR) signaling in regulating microRNA (miRNA) expression in periodontitis using wild-type, TLR2^{-/-}, and TLR4^{-/-} mouse models. The study provides valuable insights into the relationship between TLR2/4 signaling and periodontal inflammation, alveolar bone resorption (ABR), and systemic bacterial dissemination. However, several key aspects require further clarification and additional analysis to strengthen the manuscript's scientific impact. Below are detailed comments and suggestions for improvement.

Major Concerns

1. Expression of miR-146a-5p in Wild-Type and Knockout Mice:

Concern: The study reports upregulation of miR-146a-5p in both wild-type and TLR2^{-/-}/TLR4^{-/-} mice but does not explore differential expression or specific roles within each genotype.

Recommendation: I would suggest that the authors conduct pairwise comparisons of miR-146a-5p expression between wild-type and knockout groups using statistical tests such as ANOVA with post hoc Tukey's tests. This will determine whether its expression is quantitatively altered due to TLR2/4 signaling loss. Additionally, investigate whether other pattern recognition receptors (e.g., NOD-like receptors) contribute to miR-146a-5p upregulation in knockout mice.

2. Type of Knockout Model and Cell Types Implicated:

Concern: The manuscript does not specify whether the TLR2^{-/-} and TLR4^{-/-} mice are global or conditional knockouts. This distinction is critical because TLRs are expressed in diverse cell types, including gingival epithelial cells, fibroblasts, macrophages, neutrophils, and osteoclasts.

Recommendation: The authors should clarify whether the knockout models are global or conditional and specify the targeted cell types. They need to perform cell-type-specific RT-qPCR or immunohistochemistry (IHC) to determine which cells are primarily responsible for miRNA expression. Flow cytometry can be used to quantify TLR2/4 expression in immune cells and gingival tissues.

3. Pathways and Transcription Factors (TFs) Implicated:

Concern: The manuscript mentions the TLR signaling pathway but does not provide detailed insights into the transcription factors (TFs) connecting TLR activation to miRNA expression.

Recommendation: I would recommend that the authors perform Western blot or RT-qPCR to quantify key TFs such as NF- κ B (p65), IRF3, AP-1, c-Jun, STAT1, and STAT3. Investigate whether miR-146a-5p is regulated *via* NF- κ B by measuring phosphorylated NF- κ B levels and using inhibitors to assess its effect on miRNA expression. Explore alternative PRRs that may compensate for TLR deficiencies.

Minor Concerns

4. Validation of miRNA Biomarkers in Human Samples:

While the mouse model is appropriate for mechanistic insights, validation in human PD patients is essential to confirm the clinical relevance of miRNAs like miR-146a-5p and miR-15a-5p.

Suggestion: To improve the rigor of this study, the investigators need to conduct miRNA profiling using gingival crevicular fluid (GCF) or saliva samples from PD patients and healthy controls.

5. Long-Term Effects of Chronic Infection:

There is a limited time frame in this analysis. The study focuses on acute infection (10 infection cycles).

Suggestion: The long-term effects of chronic infection should be evaluated to understand the progression of PD and sustained miRNA expression changes. I would suggest extending the analysis beyond acute infection to evaluate sustained miRNA expression changes during chronic periodontitis.

6. Systemic Bacterial Dissemination:

How the bacteria disseminated was not totally skipped.

Suggestion: It is inquisitive to know how the organisms traffic systemically, especially blood-brain barrier permeability and immune evasion mechanisms that facilitate bacterial dissemination. Thus, I would suggest that the investigators should further investigate the mechanisms of bacterial translocation to systemic organs and its impact on systemic inflammation.

7. Histological Analysis:

It is disappointing to see that the authors only used basic stain tech to answer questions that have to do with various markers and proteins in tissue. The qualitative H&E staining provides limited information on inflammatory cell types.

Suggestion: The authors should use stains such as IHC to identify inflammatory cell types (macrophages, neutrophils, T cells) and quantify cytokine expression (TNF- α , IL-1 β , IL-6) in gingival tissues.

Overall Manuscript Organization and Clarity

Abstract: Pls condense the key findings and emphasize the clinical relevance of miRNA biomarkers.

Introduction: I would suggest an improvement to enhance the discussion of miRNAs' clinical significance and their potential as diagnostic tools.

Methods: Pls, clearly state the rationale for selecting the 15 miRNAs and describe the knockout models in detail.

Results: I recommend the authors should simplify figures for clarity, combining ROC curves into a single panel.

Discussion: This section is highly deficient. I recommend that the investigators provide a more comprehensive interpretation of the molecular pathways and potential therapeutic applications of identified miRNAs.

Title: Altered microRNA Expression Correlates with Reduced TLR2/4-Dependent Periodontal Inflammation and Bone Resorption Induced by Polymicrobial Infection.

Reviewers' comments.

Reviewer #1

The manuscript describes the study of miRNA expressions that are correlated with TLR2/4 that are induced by various oral bacterial treatment. The study includes a large volume of mouse samples and covers a variety of experiments. The major drawback of this manuscript is the data presentation and labelling which making the readers hard to follow. The authors have to address them before publication.

Major comments:

Comment: Figure 1C. What do 1,2,3,4,5 mean? although it seems to mentioned in line 286-291, but 1C4 and 1C4 seems not described. The authors should at least label them clearly in the figure. To compare WT and TLR2/TLR4, the authors should include all the following groups on a single graph: WT control, WT infection, TLR2 control, TLR2 infection, TLR4 control, and TLR4 infection. A two-way ANOVA statistical test should be performed to assess these comparisons. It is noted that the TLR2 and TLR4 groups show an increased ABR compared to the WT control group in the uninfected conditions. The authors need to explain if this increase is expected.

RESPONSE: As suggested by the reviewer, we have labeled Figure 1C clearly for more clarity. Figure 1C represents alveolar bone resorption (ABR) in C57BL6/J wild-type infected mice (Fig 1C1) compared to wild-type sham infection, Fig 1C2 represents ABR in TLR2^{-/-} infected mice compared to sham infection, and Fig 1C3 represent ABR in TLR4^{-/-} infected mice compared to sham infection. Fig 1C4 represents a comparison of ABR between infected wild-type mice and TLR2^{-/-} infected mice, and Fig 1C5 represents a comparison of ABR between infected wild-type mice and TLR4^{-/-} infected mice. We have included the explanations for 1C4 and 1C5 as per your suggestion. Labels were revised in Figure 1C. As suggested by the Reviewer, we have revised Fig 1C and included WT control, WT infection, TLR2^{-/-} control, TLR2^{-/-} infection, TLR4^{-/-} control, and TLR4^{-/-} infection data into a single graph (below) by performing a two-way ANOVA statistical test.

Figure 1C. Two-way ANOVA analysis of alveolar bone resorption in the wildtype, TLR2^{-/-} and TLR4^{-/-} mice.

Alveolar bone resorption (ABR) is a natural, continuous process (bone formation/resorption) in rodents that progress with age and is influenced by infection and inflammation. Our Laboratory's previous publication, Chukkapalli et al. 2016, PMID:27224005 reported that the TLR2/4 deficient mice showed ABR, where the reported ABR in TLR2^{-/-} control mice was 0.65 mm² and 0.63 mm² in TLR4^{-/-} control mice. The increased ABR in our present TLR2^{-/-} and TLR4^{-/-} control mice is also an expected result.

Comment: Fig 1D. It is unclear why error bars and statistical tests are not present.

RESPONSE: A statistical test was performed per the reviewer's suggestion. This data expresses the IgG fold change of infection over the sham. Hence, error bars cannot be shown in this presentation.

Comment: Fig 2/3/4. Why the same miRNA cannot be compared in one figure? For example: miR-146a-5p were tested in all 6 mouse samples (WT/TLR2-/TLR4-; control/infection), and all the infection samples are elevated in the miR-146a-5p.

RESPONSE: As suggested by the reviewer, we have revised the miR-146a-5p figures from Figure 2/3/4 into one figure. The miR-146a-5p was a statistically significant p-value ≤ 0.01 in all wild-types, TLR2, and TLR4 mice data analysis using the Mann-Whitney U test (Figure A). Combining all three data sets for ANOVA has a difference in the output comparative significance (Figure B). ANOVA analysis is masking the actual group difference. Accordingly, we decided to show the actual group difference between infection and sham infection in all 6 mouse samples (WT/TLR2-/TLR4-; control/infection).

A

Comment: Why ROC/AUC value are used in Fig 2/3/4 and table 5, ROC/AUC is commonly used when comparing groups in medicine/pharmacokinetic, while it is less common to compare expression in biological samples. As the statistical comparison (P-value) has been shown already, the ROC /ACU seems not providing additional information and is also not easy to be understood by general biological readers that the authors could consider to remove them.

RESPONSE: The authors thank the reviewer for the explanation. Many published research articles (a few were PMID:40011539, 40002176, 39945941, and 39836335) have used the ROC analysis for their RTqPCR data presentation. Considering its importance, we also analyzed our data for ROC analysis. Additionally, Reviewer 2 comments stated “I recommend the authors should simplify figures for clarity, combining ROC curves into a single panel”; hence, we prefer to show the ROC/AUC figures to satisfy the scientific presentation.

As suggested by Reviewer 2, the authors have made the combined ROC curves figure for upregulated miRNAs and one for the downregulated miRNAs in the wild-type infection vs sham group. Combining all the miRNAs in one figure confuses the readers and divides them into two figures.

Figure 2.

As suggested by Reviewer 2, the authors have made the combined ROC curves figure for upregulated miRNAs and downregulated miRNAs as four miRNAs in one and 3 miRNAs in one in the TLR2^{-/-} infection vs sham group. Combining all the miRNAs in one figure confuses the readers and we have divided them into three figures.

Figure 3

As suggested by Reviewer 2, the authors have made the combined ROC curves figure for upregulated miRNAs as 3 miRNAs in one, the other 3 miRNAs in one, and two miRNAs in one in the TLR4^{-/-} infection vs sham group.

Figure 4

Comment: Figure 5A-C are not easy to understand, could they be combined in one figure/alternative way to show the differences between groups?

RESPONSE: Figures 5A, 5B, and 5C are the KEGG pathways derived using differentially expressed (DE) miRNAs in wild-type mice, TLR2^{-/-} and TLR4^{-/-} mice. In our results, the DE miRNAs have differed among the wild-type, TLR2^{-/-} and TLR4^{-/-} mice. Hence, these figures could not be merged into a single figure.

Reviewer #2

General Assessment

I acknowledge the authors on their efforts investigating the role of Toll-like receptor (TLR) signaling in regulating microRNA (miRNA) expression in periodontitis using wild-type, TLR2^{-/-}, and TLR4^{-/-} mouse models. The study provides valuable insights into the relationship between TLR2/4 signaling and periodontal inflammation, alveolar bone resorption (ABR), and systemic bacterial dissemination. However, several key

aspects require further clarification and additional analysis to strengthen the manuscript's scientific impact. Below are detailed comments and suggestions for improvement.

Major Concerns

Major Concern 1: Expression of miR-146a-5p in Wild-Type and Knockout Mice:

Concern: The study reports upregulation of miR-146a-5p in both wild-type and TLR2^{-/-}/TLR4^{-/-} mice but does not explore differential expression or specific roles within each genotype.

Recommendation: I would suggest that the authors conduct pairwise comparisons of miR-146a-5p expression between wild-type and knockout groups using statistical tests such as ANOVA with post hoc Tukey's tests. This will determine whether its expression is quantitatively altered due to TLR2/4 signaling loss. Additionally, investigate whether other pattern recognition receptors (e.g., NOD-like receptors) contribute to miR-146a-5p upregulation in knockout mice.

RESPONSE: As suggested by the reviewer, we performed pairwise comparisons of miR-146a-5p expression between wild-type and knockout groups using statistical tests ANOVA. miR-146a-5p has a p-value ≤ 0.01 in the wild-type, TLR2, and TLR4 mice data analysis (Figure A) using the statistical Mann-Whitney U test. We revised by combining all the three data sets and performed the ANOVA. The ANOVA obtained results that show a difference in the comparative significance (Figure B) from the Mann-Whitney U test. We observed that the ANOVA analysis masks the actual group differences between wild-type and TLR4^{-/-} groups (Fig A).

Major Concern 2: Type of Knockout Model and Cell Types Implicated:

Concern: The manuscript does not specify whether the TLR2^{-/-} and TLR4^{-/-} mice are global or conditional knockouts. This distinction is critical because TLRs are expressed in diverse cell types, including gingival epithelial cells, fibroblasts, macrophages, neutrophils, and osteoclasts.

Recommendation: The authors should clarify whether the knockout models are global or conditional and specify the targeted cell types. They need to perform cell-type-specific RT-qPCR or immunohistochemistry (IHC) to determine which cells are primarily responsible for miRNA expression. Flow cytometry can be used to quantify TLR2/4 expression in immune cells and gingival tissues.

RESPONSE: We thank the reviewer for this valuable suggestion. The strains used in our study are B6.129-Tlr2tm1Kir/J (Strain # 004650) for TLR2 and B6(Cg)-Tlr4tm1.2Karp/J (Strain # 029015). We have incorporated these details into the Materials and Methods section (Lines 135-137). The wide literature on www.jax.org and its quoted references confirmed, and we clarify that TLR2 and TLR4 knockout mice used in our study were global knockout for TLR2 and TLR4 receptors, respectively. *Reference:* Wooten RM, Ma Y, Yoder RA, Brown JP, Weis JH, Zachary JF, Kirschning CJ, Weis JJ. Toll-like receptor 2 is required for innate, but not acquired, host defense to *Borrelia burgdorferi*. *J Immunol*. 2002 Jan 1;168(1):348-55. doi: 10.4049/jimmunol.168.1.348. PMID: 11751980. The current funded study AIMS is to determine predominant periodontal bacteria-induced global microRNAs (foundational study) and the next proposed renewal study is to examine its mechanistic role by performing cell-type-specific RT-qPCR or immunohistochemistry (IHC) to

determine which cells are primarily responsible for miRNA expression. In addition, we plan to perform flow cytometry to quantify TLR2/4 expression in immune cells and gingival tissues in future proposed studies.

Major Concern 3: Pathways and Transcription Factors (TFs) Implicated:

Concern: The manuscript mentions the TLR signaling pathway but does not provide detailed insights into the transcription factors (TFs) connecting TLR activation to miRNA expression.

Recommendation: I would recommend that the authors perform Western blot or RT-qPCR to quantify key TFs such as NF- κ B (p65), IRF3, AP-1, c-Jun, STAT1, and STAT3. Investigate whether miR-146a-5p is regulated via NF- κ B by measuring phosphorylated NF- κ B levels and using inhibitors to assess its effect on miRNA expression. Explore alternative PRRs that may compensate for TLR deficiencies.

RESPONSE: We very much value the reviewer's suggestions to examine detailed insights into the transcription factors (TFs) connecting TLR activation to miRNA expression by performing Western blot or RT-qPCR to quantify transcription factors NF- κ B (p65), IRF3, AP-1, c-Jun, STAT1, and STAT3. As stated above, the current funded study AIMS is to determine predominant periodontal bacteria-induced global microRNAs (foundational study) and the next proposed renewal study will examine transcription factors (TFs) connecting TLR activation to miRNA expression. Similarly, we will examine whether miR-146a-5p is regulated via NF- κ B by measuring phosphorylated NF- κ B levels by using inhibitors to assess its effect on miRNA expression. In addition, as suggested we will explore alternative PRRs that may compensate for TLR deficiencies in our future studies.

Minor Concerns

Minor Concern 1: Validation of miRNA Biomarkers in Human Samples:

While the mouse model is appropriate for mechanistic insights, validation in human PD patients is essential to confirm the clinical relevance of miRNAs like miR-146a-5p and miR-15a-5p.

Suggestion: To improve the rigor of this study, the investigators need to conduct miRNA profiling using gingival crevicular fluid (GCF) or saliva samples from PD patients and healthy controls.

RESPONSE: We strongly agree with the reviewer's comment that in order for miRNA biomarkers to be relevant in human PD patients, extensive analysis is essential with large and appropriate cohorts. Of course, this is beyond the scope of the present study.

Minor Concern 2: Long-Term Effects of Chronic Infection: There is a limited time frame in this analysis. The study focuses on acute infection (10 infection cycles).

Suggestion: The long-term effects of chronic infection should be evaluated to understand the progression of PD and sustained miRNA expression changes. I would suggest extending the analysis beyond acute infection to evaluate sustained miRNA expression changes during chronic periodontitis.

RESPONSE: We agree with the reviewer's suggestion to extend our studies to examine long-term effects of chronic infection to understand the progression of PD and sustained miRNA altered expression compared to current acute infection. We will include chronic infection studies in our planned new proposal.

Minor Concern 3: Systemic Bacterial Dissemination: How the bacteria disseminated was not totally skipped.

Suggestion: It is inquisitive to know how the organisms traffic systemically, especially blood-brain barrier permeability and immune evasion mechanisms that facilitate bacterial dissemination. Thus, I would suggest that the investigators should further investigate the mechanisms of bacterial translocation to systemic organs and its impact on systemic inflammation.

RESPONSE: We value the reviewer's suggestion. There has been no mechanistic definition of how subgingival bacteria can disseminate, what cell types are targeted by oral bacteria during infection, and whether disseminated bacteria are transported in circulating blood cells to the arterial wall. To the best of our knowledge, no investigations have identified the cell types (RBCs, neutrophils, monocytes, dendritic cells) targeted for the transport of periodontal bacteria and nor have investigations demonstrated whether multiple oral bacterial species or single bacteria can infect individual cells by using fluorescent-protein labeled oral bacteria to track the cell types. We will also propose to demonstrate dissemination of oral bacteria in mouse models with

neutrophil-depleted, neutrophil elastase (NE) deficient, myeloperoxidase (MPO) deficient mice or macrophage-depleted mice in our future studies.

Minor Concern 4: Histological Analysis: It is disappointing to see that the authors only used basic stain tech to answer questions that have to do with various markers and proteins in tissue. The qualitative H&E staining provides limited information on inflammatory cell types.

Suggestion: The authors should use stains such as IHC to identify inflammatory cell types (macrophages, neutrophils, T cells) and quantify cytokine expression (TNF- α , IL-1 β , IL-6) in gingival tissues.

RESPONSE: In this study, our major focus is on microRNA expression to polymicrobial infection in three different models including TLR2/4 mice and we routinely examine PD major outcome measures (gingival inflammation, and ABR). We have labelled H & E figures for clarity. However, as the reviewer suggested we will use IHC to identify gingival inflammatory cell types (macrophages, neutrophils, T cells) and will quantify cytokine expression (TNF- α , IL-1 β , IL-6) in mouse gingival tissues in future studies.

Overall Manuscript Organization and Clarity

Abstract: Pls condense the key findings and emphasize the clinical relevance of miRNA biomarkers.

RESPONSE: We have revised the abstract to emphasize the clinical relevance of miRNA biomarkers.

Introduction: I would suggest an improvement to enhance the discussion of miRNAs' clinical significance and their potential as diagnostic tools.

RESPONSE: We agree with the reviewer's suggestions. In the introduction, we have included the discussion of miRNAs' clinical significance and their potential as diagnostic tools (Lines # 67 to 73). Two references related to miRNAs' clinical significance and potential as diagnostic tools were included in the revised manuscript.

Methods: Pls, clearly state the rationale for selecting the 15 miRNAs and describe the knockout models in detail.

RESPONSE: We have provided the rationale for selecting the 15 miRNAs (Lines # 124 – 127) in the introduction section. The rationale for selecting the 15 miRNAs was that the preclinical *in vivo* studies involving five different bacteria (*S. gordonii*, *F. nucleatum*, *P. gingivalis*, *T. denticola*, *T. forsythia*) have revealed specific differentially expressed (DE) miRNAs in each infection that was analyzed using NanoString nCounter microRNA Expression technology. These miRNAs exhibit a unique range of expression patterns during polymicrobial infections (*S. gordonii*/*P. gingivalis*/*F. nucleatum*/*T. denticola*/*T. forsythia*) as well as five distinct monoinfection (*S. gordonii*, *P. gingivalis*, *F. nucleatum*, *T. denticola*, *T. forsythia*) with the highest upregulated and downregulated miRNAs with high fold change in 15 miRNAs that were selected for investigation of its presence in C57BL6/J wild-type, TLR2^{-/-}, and TLR4^{-/-} gene knockout mice (Table 2, Columns 4, 5) mandibles to polymicrobial infection. The 4th and 5th columns subdividing upregulated and downregulated in Table 2 also explained the 15 miRNAs selection criteria.

We have described the information on TLR knockout mice in abstract (Lines # 24 and 25) and the introduction (Lines # 123 and 124). Additionally, the Materials and Methods section explains that “this experiment used C57BL6/J wild-type, TLR2^{-/-} B6.129-Tlr2tm1Kir/J (Strain # 004650) a global knockout for TLR2 receptors, and TLR4^{-/-} B6(Cg)-Tlr4tm1.2Karp/J (Strain # 029015) mice a global knockout for TLR4 receptors were purchased from Jackson Laboratory (Bar Harbor, ME, USA)”.

Results: I recommend the authors should simplify figures for clarity, combining ROC curves into a single panel.

RESPONSE: We thank the reviewer for the valuable suggestion. We combined the ROC curves into a single panel for up and downregulated microRNAs. ROC curves combined for upregulated and downregulated miRNAs as one in the wild-type infection vs sham group. Combining all the miRNAs in one figure will confuse the readers and we have divided them into two figures (below).

Figure 2

ROC curves combined for upregulated miRNAs as one and downregulated miRNAs as four miRNAs in one and 3 miRNAs in one in the TLR2^{-/-} infection vs sham group. Combining all the miRNAs in one figure will confuse the readers and we have divided them into three figures (below).

Figure 3

ROC curves combined for upregulated miRNAs as 3 miRNAs in one, 3 miRNAs in one, and two miRNAs in one in the TLR4^{-/-} infection vs sham group. Combining all the miRNAs in one figure will confuse the readers and we have divided them into three figures (below).

Figure 4

Discussion: This section is highly deficient. I recommend that the investigators provide a more comprehensive interpretation of the molecular pathways and potential therapeutic applications of identified miRNAs.

RESPONSE: We thank the reviewer for the valuable suggestion. We have revised the discussion section in the manuscript text. We explained the therapeutic applications of our experiment and identified miRNAs in lines # 641 – 642; 666-672. Additionally, Tabel 5 has detailed descriptions of the target functions of each upregulated miRNA, which explains the therapeutic applications in clinical studies.

Re: Spectrum00160-25R1 (Altered microRNA Expression Correlates with Reduced TLR2/4-Dependent Periodontal Inflammation and Bone Resorption Induced by Polymicrobial Infection)

Dear Dr. Kesavalu Lakshmyya:

Thank you for the privilege of reviewing your work. Below you will find my comments, instructions from the Spectrum editorial office, and the reviewer comments.

Please see the referees' comments below that have not been addressed appropriately by the authors. Please revise and take the following comments into consideration in the revised version of the manuscript:

1) "Comment: Fig 1D. It is unclear why error bars and statistical tests are not present. RESPONSE: A statistical test was performed per the reviewer's suggestion. This data expresses the IgG fold change of infection over the sham. Hence, error bars cannot be shown in this presentation."

Lines 306-307 (figure legend Fig 1D): The authors responded to the referees that "This data expresses the IgG fold change of infection over the sham. Hence, error bars cannot be shown in this presentation.". However, the revised version of the manuscript submitted to this journal shows the following sentence: "(D) Bacteria-specific IgG antibody analysis. Ordinary two-way ANOVA). Data points and error bars are mean{plus minus} SEM (n = 16). Please correct the written portion of the manuscript or correct the graphs (including error bars and the control group).

2) "Comment: Fig 2/3/4. Why the same miRNA cannot be compared in one figure? For example: miR-146a-5p were tested in all 6 mouse samples (WT/TLR2-/TLR4-; control/infection), and all the infection samples are elevated in the miR-146a-5p.

RESPONSE: As suggested by the reviewer, we have revised the miR-146a-5p figures from Figure 2/3/4 into one figure. The miR-146a-5p was a statistically significant p-value {less than or equal to}0.01 in all wild-types, TLR2, and TLR4 mice data analysis using the Mann-Whitney U test (Figure A). Combining all three data sets for ANOVA has a difference in the output comparative significance (Figure B). ANOVA analysis is masking the actual group difference. Accordingly, we decided to show the actual group difference between infection and sham infection in all 6 mouse samples (WT/TLR2-/TLR4-; control/infection)."

"Concern: The study reports upregulation of miR-146a-5p in both wild-type and TLR2-/-/TLR4-/- mice but does not explore differential expression or specific roles within each genotype.

Recommendation: I would suggest that the authors conduct pairwise comparisons of miR-146a-5p expression between wild-type and knockout groups using statistical tests such as ANOVA with post hoc Tukey's tests. This will determine whether its expression is quantitatively altered due to TLR2/4 signaling loss. Additionally, investigate whether other pattern recognition receptors (e.g., NOD-like receptors) contribute to miR-146a-5p upregulation in knockout mice."

Both referees suggested that the authors combine the results for miR-146a-5p for all 6 mouse (between WT and knockouts) samples tested in one graph and perform using statistical tests such as ANOVA. The authors must add this graph to the manuscript and run the appropriate statistical analyses, as suggested by the referees. The authors must also add any additional comments on the statistical results to the "discussion" or "results" sections.

3) "Recommendation: I would recommend that the authors perform Western blot or RT-qPCR to quantify key TFs such as NF- κ B (p65), IRF3, AP-1, c-Jun, STAT1, and STAT3. Investigate whether miR-146a-5p is regulated via NF- κ B by measuring phosphorylated NF- κ B levels and using inhibitors to assess its effect on miRNA expression. Explore alternative PRRs that may compensate for TLR deficiencies.

RESPONSE: We very much value the reviewer's suggestions to examine detailed insights into the transcription factors (TFs) connecting TLR activation to miRNA expression by performing Western blot or RT-qPCR to quantify transcription factors NF- κ B (p65), IRF3, AP-1, c-Jun, STAT1, and STAT3. As stated above, the current funded study AIMS is to determine predominant periodontal bacteria-induced global microRNAs (foundational study) and the next proposed renewal study will examine transcription factors (TFs) connecting TLR activation to miRNA expression. Similarly, we will examine whether miR-146a-5p is regulated via NF- κ B by measuring phosphorylated NF- κ B levels by using inhibitors to assess its effect on miRNA expression. In addition, as suggested we will explore alternative PRRs that may compensate for TLR deficiencies in our future studies."

Since the authors did not include the suggestion by the referee, the authors must include, at least, a discussion of the potential

implications of investigating key transcription factors involved in TLRs activation.

4) "Minor Concern 1: Validation of miRNA Biomarkers in Human Samples: While the mouse model is appropriate for mechanistic insights, validation in human PD patients is essential to confirm the clinical relevance of miRNAs like miR-146a-5p and miR-15a-5p. Suggestion: To improve the rigor of this study, the investigators need to conduct miRNA profiling using gingival crevicular fluid (GCF) or saliva samples from PD patients and healthy controls.

RESPONSE: We strongly agree with the reviewer's comment that in order for miRNA biomarkers to be relevant in human PD patients, extensive analysis is essential with large and appropriate cohorts. Of course, this is beyond the scope of the present study."

Since the authors did not include the suggestion by the referee, the authors must include, at least, a discussion of the importance of conducting experiments with human samples and the potential implications of their study to understand human diseases.

5) "Minor Concern 3: Systemic Bacterial Dissemination: How the bacteria disseminated was not totally skipped. Suggestion: It is inquisitive to know how the organisms traffic systemically, especially blood-brain barrier permeability and immune evasion mechanisms that facilitate bacterial dissemination. Thus, I would suggest that the investigators should further investigate the mechanisms of bacterial translocation to systemic organs and its impact on systemic inflammation.

RESPONSE: We value the reviewer's suggestion. There has been no mechanistic definition of how subgingival bacteria can disseminate, what cell types are targeted by oral bacteria during infection, and whether disseminated bacteria are transported in circulating blood cells to the arterial wall. To the best of our knowledge, no investigations have identified the cell types (RBCs, neutrophils, monocytes, dendritic cells) targeted for the transport of periodontal bacteria and nor have investigations demonstrated whether multiple oral bacterial species or single bacteria can infect individual cells by using fluorescent-protein labeled oral bacteria to track the cell types. We will also propose to demonstrate dissemination of oral bacteria in mouse models with neutrophil-depleted, neutrophil elastase (NE) deficient, myeloperoxidase (MPO) deficient mice or macrophage-depleted mice in our future studies."

Authors must include at least a discussion (and expand the current discussion) on how they believe or hypothesize that bacteria spread to different sites and tissues of the mouse post in vivo infection.

Revision Guidelines

Data availability: ASM policy requires that data be available to the public upon online posting of the article, so please verify all links to sequence records, if present, and make sure that each number retrieves the full record of the data. If a new accession number is not linked or a link is broken, provide Spectrum production staff with the correct URL for the record. If the accession

numbers for new data are not publicly accessible before the expected online posting of the article, publication may be delayed; please contact production staff (Spectrum@asmusa.org) immediately with the expected release date.

Sincerely,
Cassio Almeida-da-Silva
Editor
Microbiology Spectrum

Reviewer Comments

1) "Comment: Fig 1D. It is unclear why error bars and statistical tests are not present.

RESPONSE: A statistical test was performed per the reviewer's suggestion. This data expresses the IgG fold change of infection over the sham. Hence, error bars cannot be shown in this presentation."

Lines 306-307 (figure legend Fig 1D): The authors responded to the referees that "This data expresses the IgG fold change of infection over the sham. Hence, error bars cannot be shown in this presentation.". However, the revised version of the manuscript submitted to this journal shows the following sentence: "(D) Bacteria specific IgG antibody analysis. Ordinary two-way ANOVA). Data points and error bars are mean \pm SEM (n = 16). Please correct the written portion of the manuscript or correct the graphs (including error bars and the control group).

Response: As suggested by the reviewers, the written portion of the manuscript was corrected from lines 290 to 298, as shown here "Bacteria-specific IgG antibody shown in Figure 1D was a presentation of IgG fold change in the infection over the sham. The analysis was performed using Ordinary two-way ANOVA, and the data shown in fold-change of IgG levels in the infection over sham (n = 16). A significant increase of IgG immune response was observed in infected mice groups; specifically the wild-type mice exhibited highly significant serum IgG response >1000 folds to *P. gingivalis* (p < 0.0001) and *T. forsythia* (p < 0.0001), TLR2^{-/-} and TLR4^{-/-} mice showed significant IgG response to *F. nucleatum* (p < 0.0001) and *P. gingivalis* (p < 0.0001) (>90 folds), and TLR4^{-/-} mice had a robust level of IgG response to *P. gingivalis* (p < 0.01)(>700 folds) (Figure 1D1-3)." Similar edits were made in the Figure legend 1 (line # 986 to 990).

2) "Comment: Fig 2/3/4. Why the same miRNA cannot be compared in one figure? For example: miR-146a-5p were tested in all 6 mouse samples (WT/TLR2^{-/-}/TLR4^{-/-}; control/infection), and all the infection samples are elevated in the miR-146a-5p.

"Concern: The study reports upregulation of miR-146a-5p in both wild-type and TLR2^{-/-}/TLR4^{-/-} mice but does not explore differential expression or specific roles within each genotype.

Recommendation: I would suggest that the authors conduct pairwise comparisons of miR-146a-5p expression between wild-type and knockout groups using statistical tests such as ANOVA with post hoc Tukey's tests. This will determine whether its expression is quantitatively altered due to TLR2/4 signaling loss. Additionally, investigate whether other pattern recognition receptors (e.g., NOD-like receptors) contribute to miR-146a-5p upregulation in knockout mice."

Both referees suggested that the authors combine the results for miR-146a-5p for all 6 mice

(between WT and knockouts) samples tested in one graph and perform using statistical tests such as ANOVA. The authors must add this graph to the manuscript and run the appropriate statistical analyses, as suggested by the referees. The authors must also add any additional comments on the statistical results to the "discussion" or "results" sections.

Response: As suggested by both referees, we performed a statistical test, ANOVA with post hoc Tukey's test for miR-146a-5p to all 6 mice (between WT and knockout) samples and presented in Figure 5 located in line # 409. Statistical results shown in results section, lines 390 to 392. In discussion section (line# 496-498) we stated “miR-146a is a master regulator involved in controlling multiple TLR signaling pathways, including TLR4/2, but also other TLRs [96,97]”.

Figure 5.

3) "Recommendation: I would recommend that the authors perform Western blot or RT-qPCR to quantify key TFs such as NF-κB (p65), IRF3, AP-1, c-Jun, STAT1, and STAT3. Investigate whether miR-146a-5p is regulated via NF-κB by measuring phosphorylated NF-κB levels and using inhibitors to assess its effect on miRNA expression. Explore alternative PRRs that may compensate for TLR deficiencies.

Since the authors did not include the suggestion by the referee, the authors must include, at least, a discussion of the potential implications of investigating key transcription factors involved in TLRs activation.

Response: We very much value the reviewer's suggestions to examine detailed insights into the transcription factors (TFs) connecting TLR activation to miRNA expression by performing Western blot or RT-qPCR to quantify transcription factors NF-κB (p65), IRF3, AP-1, c-Jun, STAT1, and STAT3. As suggested by the reviewers, we have included in discussion Line 574 to 588, as shown “*Treponema denticola* virulence factors activate TLR2 (PMID: 39956576), and *S. aureus* lipoprotein stimulate NF-κB pathway (PMID: 18191935). Bone marrow derived macrophages respond to *P. gingivalis* with the activation and nuclear translocation of interferon

regulatory factor 3 (IRF3) (PMID: 23803413), human monocytes infected with gingipains induced c-Jun/c-Fos (AP-1) and phosphorylated NF- κ B (PMID: 35110422) and signal transducer and activator of transcription (STAT 1/STAT3). Quantifying these key transcription factors (TFs) and exploring the alternative PRRs that may compensate for TLR deficiencies could have potential implications in elucidating the molecular components involved in TLR activation.”

4) "Minor Concern 1: Validation of miRNA Biomarkers in Human Samples: While the mouse model is appropriate for mechanistic insights, validation in human PD patients is essential to confirm the clinical relevance of miRNAs like miR-146a-5p and miR-15a-5p. Suggestion: To improve the rigor of this study, the investigators need to conduct miRNA profiling using gingival crevicular fluid (GCF) or saliva samples from PD patients and healthy controls. Since the authors did not include the suggestion by the referee, the authors must include, at least, a discussion of the importance of conducting experiments with human samples and the potential implications of their study to understand human diseases.

Response: We strongly agree with the reviewer's comment that for miRNA biomarkers to be relevant in human PD patients, extensive analysis is essential with large and appropriate cohorts. As suggested by the reviewers, we have included in discussion, lines 586 to 588, as shown “We acknowledge that the result from our current study is based on an excellent murine model of PD, but, in order for this work to show impact in human disease, it is logical to plan detail studies with the human disease”.

5) "Minor Concern 3: Systemic Bacterial Dissemination: How the bacteria disseminated was not totally skipped. Suggestion: It is inquisitive to know how the organisms traffic systemically, especially blood-brain barrier permeability and immune evasion mechanisms that facilitate bacterial dissemination. Thus, I would suggest that the investigators should further investigate the mechanisms of bacterial translocation to systemic organs and its impact on systemic inflammation.

Authors must include at least a discussion (and expand the current discussion) on how they believe or hypothesize that bacteria spread to different sites and tissues of the mouse post in vivo infection.

Response: We agree with the reviewer comment. As suggested by the reviewers, we have included in discussion, line # 574 to 578 as shown “We hypothesize that during severe periodontitis, subgingival periodontal bacteria might enter directly into the bloodstream through the subgingival ulcerated gingival epithelium, causing bacterial translocation to distant organs, and colonize in multiple organs [104]. Future studies will direct to investigate the mechanisms of oral bacterial translocation to multiple systemic organs (heart, lungs, liver, spleen, kidney, and brain) and its impact on systemic inflammation.”

Re: Spectrum00160-25R2 (Altered microRNA Expression Correlates with Reduced TLR2/4-Dependent Periodontal Inflammation and Bone Resorption Induced by Polymicrobial Infection)

Dear Dr. Kesavalu Lakshmyya:

Your manuscript has been accepted, and I am forwarding it to the ASM production staff for publication. Your paper will first be checked to make sure all elements meet the technical requirements. ASM staff will contact you if anything needs to be revised before copyediting and production can begin. Otherwise, you will be notified when your proofs are ready to be viewed.

Sincerely,
Cassio Almeida-da-Silva
Editor
Microbiology Spectrum